# An RNA-based feed-forward mechanism ensures motor switching in *oskar* mRNA transport

Imre Gáspár[1] , Ly Jane Phea[1] , Mark A. McClintock[2] , Simone Heber[1] , Simon L. Bullock[2] , and Anne Ephrussi[1]

**Regulated recruitment and activity of motor proteins is essential for intracellular transport of cargoes, including messenger ribonucleoprotein complexes (RNPs). Here, we show that orchestration of *oskar* RNP transport in the *Drosophila* germline relies on interplay between two double-stranded RNA-binding proteins, Staufen and the dynein adaptor Egalitarian (Egl). We find that Staufen antagonizes Egl-mediated transport of *oskar* mRNA by dynein both in vitro and in vivo. Following delivery of nurse cell-synthesized *oskar* mRNA into the oocyte by dynein, recruitment of Staufen to the RNPs results in dissociation of Egl and a switch to kinesin-1-mediated translocation of the mRNA to its final destination at the posterior pole of the oocyte. We additionally show that Egl associates with *staufen* (*stau*) mRNA in the nurse cells, mediating its enrichment and translation in the ooplasm. Our observations identify a novel feed-forward mechanism, whereby dynein-dependent accumulation of *stau* mRNA, and thus protein, in the oocyte enables motor switching on *oskar* RNPs by downregulating dynein activity.**

## Introduction

Proper execution of the genetic program relies on regulatory mechanisms that act at multiple levels of gene expression, from transcription to post-translational events. In many cell types, including neurons, early embryos, and oocytes, a key process that regulates genetic output at the subcellular level is RNA localization, which is often driven by cytoskeletal motors (St Johnston, 2005; Gaspar, 2011; Marchand et al. 2012; Glock et al. 2017; Mofatteh and Bullock, 2017). The kinesin-1 and cytoplasmic dynein-1 (dynein) motors, which move RNPs toward the plus end and minus end of microtubules, respectively, play a central role in mRNA trafficking in several systems (Gaspar, 2011; Mofatteh and Bullock, 2017).

A paradigm for the study of microtubule-based RNP transport is *oskar* mRNA, the protein product of which induces abdomen and germline formation in the embryo (Ephrussi and Lehmann, 1992). *oskar* mRNA is produced in the nurse cells of the *Drosophila* germline syncytium in the early stages of oogenesis and transported by dynein into the developing oocyte where microtubule minus ends are enriched (Ephrussi et al. 1991; Januschke et al., 2002; Clark et al. 2007; Sanghavi et al., 2013; Jambor et al., 2014; Vazquez-Pianzola et al., 2017). mRNA trafficking from the nurse cells to the oocyte is dependent on the Egl protein, which binds specialized stem-loop structures in localizing mRNAs and links them, together with its coiled-coil-containing binding partner Bicaudal-D (BicD), to dynein and the

dynein activating complex dynactin (Navarro et al., 2004; Dienstbier et al., 2009; Dix et al., 2013). In vitro reconstitution studies have shown that Egl and BicD also play a key role in switching on dynein motility (McClintock et al., 2018; Sladewski et al., 2018).

During mid-oogenesis, when the microtubule cytoskeleton is reorganized, most of Egl's mRNA targets localize to the anterior of the oocyte, where microtubule minus ends are now concentrated (Gaspar, 2011; Lasko, 2012). In contrast, *oskar* mRNA is transported to the posterior pole, where the microtubule plus ends are focused (Parton et al., 2011; Sanghavi et al., 2013), by the Kinesin-1 heavy chain (Khc; Brendza et al., 2000; Palacios and St Johnston, 2002; Zimyanin et al., 2008; Williams et al., 2014). Khc is associated with nascent *oskar* RNPs in the nurse cell cytoplasm (Gáspár et al., 2017), raising the question of how the activity of the motor is coordinated with that of dynein during delivery of *oskar* mRNA from the nurse cells to the oocyte posterior.

Another double-stranded RNA-binding protein (dsRBP), Staufen, is necessary for the Khc-mediated transport of *oskar* to the posterior pole (Ephrussi et al. 1991; St Johnston et al. 1991; Januschke et al., 2002; Sanghavi et al., 2013; Jambor et al., 2014; Vazquez-Pianzola et al., 2017; St Johnston et al., 1992; Zimyanin et al., 2008), as well as for production of Oskar protein at this site (St Johnston et al. 1991; Ephrussi and Lehmann, 1992; Schuldt et al., 1998; Micklem et al., 2000). Staufen is also involved in

[1]Developmental Biology Unit, European Molecular Biology Laboratory, Heidelberg, Germany; [2]Division of Cell Biology, MRC Laboratory of Molecular Biology, Cambridge, UK.

Correspondence to Imre Gáspár: imre.gaspar@turbine.ai; Simon L. Bullock: sbullock@mrc-lmb.cam.ac.uk; Anne Ephrussi: anne.ephrussi@embl.org

I. Gáspár's current affiliation is Turbine.ai, Budapest, Hungary.

trafficking of other mRNAs, such as *bicoid* in the oocyte and *prospero* in dividing neuroblasts (Ferrandon et al., 1994; Schuldt et al., 1998). Mammalian orthologs of *Drosophila* Staufen, mStau1 and mStau2, play roles in the bidirectional transport of *CaMKIIα* and *Rgs4* mRNAs in dendrites (Heraud-Farlow et al., 2013; Bauer et al., 2019), and *Xenopus* Staufen is a component of *Vg1* transport RNPs in the oocyte (Yoon and Mowry, 2004). These observations indicate that Staufen's function as a regulator of RNP transport is evolutionarily conserved. However, it is unclear how Staufen proteins orchestrate mRNA-trafficking processes.

Here, we show that Staufen antagonizes dynein-based transport of *oskar* RNPs within the *Drosophila* oocyte and thus favors kinesin-1-driven translocation of the mRNA to the posterior pole. Recombinant Staufen can inhibit dynein-based transport of *oskar* mRNPs that are reconstituted in vitro with purified components, showing that it is a direct regulator of the transport machinery. Staufen displaces Egl from *oskar* RNPs after they have reached the oocyte. Staufen does not, however, perturb dynein and dynactin association with the RNPs, revealing that proteins in addition to Egl contribute to motor linkage in this system. These data suggest that Staufen interferes with dynein activation by promoting dissociation of Egl from RNPs. We also provide evidence that Staufen's inhibition of dynein is restricted to the oocyte by Egl- and dynein-mediated concentration of nurse cell-synthesized *stau* mRNA within the ooplasm. Thus, Egl delivers its own negative regulator into the oocyte, constituting a feed-forward regulatory mechanism that spatially and temporally controls dynein activity and thus *oskar* mRNA localization during oogenesis.

## Results

### Staufen suppresses minus-end-directed transport of *oskar* RNPs in ooplasmic extracts

*oskar* mRNA is localized at the posterior pole of the oocyte during mid-oogenesis (Fig. 1 A [Ephrussi et al. 1991; Kim-Ha et al. 1991]). In *stau* mutant oocytes, enrichment of *oskar* mRNA at the posterior pole is greatly reduced, and a considerable amount of *oskar* remains at the anterior margin (Fig. 1 A' and Fig. S1, A–C [Ephrussi et al. 1991; St Johnston et al. 1991]). An almost identical *oskar* mislocalization phenotype was observed in *stau* RNAi-expressing oocytes (Fig. S1, B and C). To assess the effect of *stau* depletion on the balance of *oskar* directionality on microtubules, we evaluated the movement of individual MCP-GFP-labeled *oskar-MS2* RNPs in control and *stau* RNAi ooplasmic extracts containing the microtubule plus end marker EB1-mCherry by TIRF microscopy (Gaspar and Ephrussi, 2017; Gáspár et al., 2017). This analysis revealed that the normal ~2:1 dominance of microtubule plus-end- versus minus-end-directed *oskar* RNP runs (Gaspar and Ephrussi, 2017; Gáspár et al., 2017) was lost in *stau* RNAi extracts, such that there was no overt bias in the directionality of motility (Fig. 1 B, two left-hand columns).

While analyzing the motility, we noticed that the relative GFP intensity of the motile fraction of *oskar* RNPs in the *stau* RNAi extracts was lower than in the control RNAi condition, indicating fewer mRNA molecules on average. This observation might explain the paucity of *oskar* mRNA transport detected in

live *stau* mutant oocytes when using the less-sensitive confocal imaging technique (compare Videos 1, 2, 3, and 4 [Zimyanin et al., 2008]). However, loss of Staufen does not seem to affect the overall RNA content of *oskar* RNPs in the oocyte as revealed by single-molecule fluorescent in situ hybridization (smFISH; Fig. S1 D). These observations indicate a selective effect of Staufen depletion on the size of motile RNPs.

Since differences in cargo size could confound the analysis of motile RNP behavior, we compared movement of *oskar* RNPs of similar sizes by stratifying the GFP intensity of RNPs into "relative" RNA units (see Materials and methods and Fig. 1 C). This approach confirmed that motile *oskar* RNPs contained fewer RNA molecules on average in the *stau* RNAi condition (Fig. 1 C) and additionally showed that the loss of plus end dominance could be observed for motile RNPs with 1 or 2 relative units of *oskar* mRNA, which make up the majority of the moving population (Fig. 1 B, four central columns). These data show that Staufen is needed for the plus end dominance that localizes *oskar* to the posterior pole.

The distribution of *oskar* mRNA and the directionality of RNP transport in *stau* mutants differ from what is observed in other mutants affecting *oskar* localization: in *Tm1$^{gs}$* (Fig. 1 A''; Gáspár et al., 2017) and *Khc$^{null}$* (Brendza et al., 2000; Zimyanin et al., 2008; Williams et al., 2014; Gáspár et al., 2017) mutants, *oskar* mRNA does not localize at the posterior of the oocyte, and minus-end-directed transport predominates because of a failure of Khc-mediated posterior-ward transport (Gáspár et al., 2017). To explore the basis of the directionality defect in *stau* mutants, we quantified the colocalization in egg chambers of functional, endogenously tagged fluorescent Khc molecules and *oskar* mRNA detected by smFISH. This method was previously used to show that Tm1 mutations reduce the association of *oskar* RNPs with the motor (Gáspár et al., 2017). When Staufen was depleted, there was no reduction in Khc colocalization with *oskar* mRNA in either the nurse cells or the oocyte (Fig. 1 D), indicating that the observed loss of plus-end-directed bias (Fig. 1 B) is not due to reduced Khc association with *oskar* RNPs.

To assess whether there is a change in motor activity when Staufen is disrupted, we analyzed the speed and run lengths of GFP-labeled *oskar-MS2* RNPs toward both microtubule ends in *stau* RNAi and control ooplasmic extracts. The velocity of *oskar* RNP movement appears to be influenced by their RNA content, as larger RNPs generally moved more slowly in either direction (Fig. 1 E). When Staufen was depleted, there was a substantial and statistically significant increase in minus-end-directed run lengths and velocities of RNPs containing one or two relative units of *oskar* mRNA (Fig. 1, E and F). In contrast, features of plus-end-directed motion were generally not affected in the *stau* RNAi condition with the exception of a modest, but significant, increase in plus-end-directed velocity for RNPs with 2 *oskar* mRNA units (Fig. 1 E). These data indicate that the loss of plus-end-directed dominance of *oskar* RNPs when Staufen is depleted is predominantly associated with hyperactivity of the minus-end-directed dynein transport machinery. Previous work with gain-of-function mutations in the dynein activator BicD showed that increased dynein activity leads to ectopic anterior accumulation of *oskar* mRNA (Navarro et al., 2004; Liu et al., 2013), as

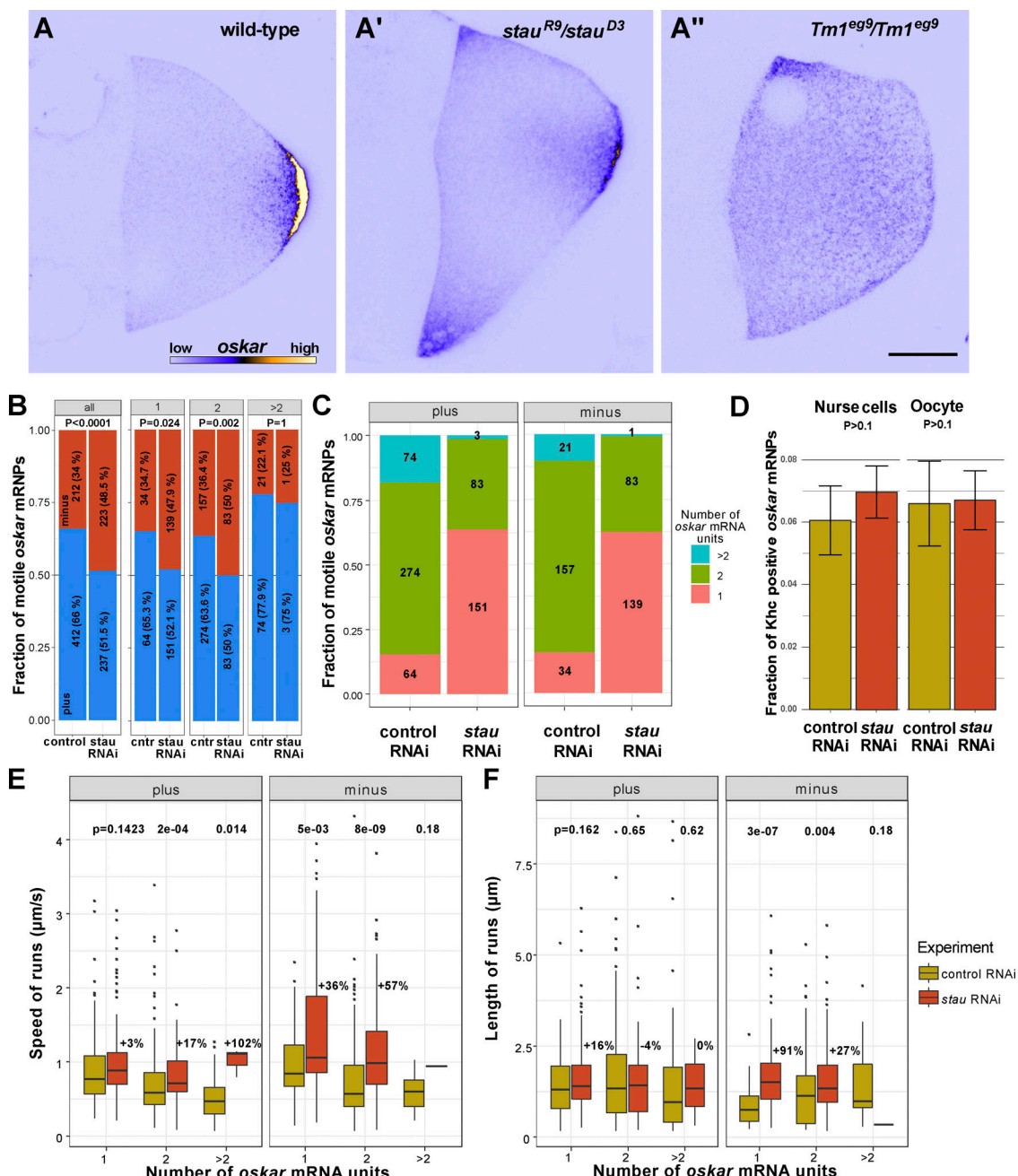

Figure 1. **oskar mRNA localization and transport are impaired in stau mutants. (A–A")** Endogenous oskar mRNA localization (shown in blue [low intensity RNA signal]/yellow [high intensity RNA signal]) in wild-type (A), stau null (A'), and Tm1-I/C null stage 9 oocytes (A"; anterior to the left and posterior to the right). Scale bar represents 20 µm. In stau null oocytes, there is a pronounced anterior accumulation of oskar in addition to the normal posterior localization; in Tm1-I/C mutant oocytes, oskar mRNA is predominantly enriched at the anterior cortex. **(B)** Polarity distribution of oskar-MS2 RNP runs in control RNAi and stau RNAi ooplasmic extracts. Numbers indicate the number of oskar RNP runs measured for all motile oskar RNPs and for RNPs categorized by relative RNA content (gray boxes, see Materials and methods; Video 1 and 2). **(C)** Relative mRNA content of motile oskar-MS2 RNPs in control RNAi and stau RNAi ooplasmic extracts. Pink, green, and blue indicate the fraction of oskar RNPs with one, two, or more relative RNA units, respectively (see Materials and methods). Numbers show the number of oskar RNPs in each category. **(D)** Association of Khc and oskar RNPs in situ in the nurse cells and the oocyte of control RNAi versus stau RNAi samples. The number of analyzed oskar RNPs was 15,506 (control, nurse cells), 14,409 (stau RNAi, nurse cells), 7,973 (control, oocytes), and 11,183 (stau RNAi, oocytes). **(E and F)** Speed (E) and run length (F) of motile oskar-MS2 RNPs toward the plus or minus ends of microtubules in control RNAi and stau RNAi ooplasmic extracts. RNPs are stratified by relative RNA content (C). Percentage values above boxplots (E and F) indicate differences in mean travel distance between stau and control RNAi extracts. In stau RNAi extracts there were too few motile oskar-MS2 RNPs with >2 RNA units to establish a statistical trend (C, E, and F). P-values of Fisher exact (B) or pairwise Mann–Whitney U-tests (D–F) are shown.

is observed in *stau* mutant oocytes (Fig. 1 A′; St Johnston et al., 1991). Altogether, these findings point to a role of Staufen in suppression of minus-end-directed transport of *oskar* mRNA by dynein.

## Staufen antagonizes dynein-mediated *oskar* mRNA transport by purified components

To test if Staufen has a direct inhibitory effect on minus-end-directed *oskar* RNP transport, we made use of an in vitro motility assay that involves reconstitution of a minimal RNP assembled from purified Egl, BicD, dynactin, dynein and in vitro transcribed RNA (McClintock et al., 2018). Motility of this RNP along immobilized, fluorescent microtubules can be visualized by TIRF microscopy using fluorophores added to the mRNA and dynein.

In these experiments we used a ∼529 nt fragment of the *oskar* 3′UTR that is sufficient for nurse cell-to-oocyte transport (Jambor et al., 2014). In the presence of Egl, BicD and dynactin, *oskar* mRNA frequently underwent long-distance transport in association with dynein (Fig. 2 A and Fig. S2 A). No transport of *oskar* was detected when Egl and BicD were omitted from the mix (Fig. S2 A); in this condition, we almost never detected association of *oskar* mRNA with microtubules or processive movement of microtubule-bound dynein complexes (Fig. S2, A–C). These observations corroborate the conclusion that Egl and BicD link mRNAs to dynein-dynactin and activate movement of the motor along microtubules (Dienstbier et al., 2009; McClintock et al., 2018; Sladewski et al., 2018).

Addition of recombinant Staufen to the Egl, BicD, dynein and dynactin assembly mix significantly reduced the number of *oskar* mRNA transport events (Fig. 2, A and B). This effect was associated with fewer mRNAs being recruited to the microtubules (Fig. 2 C). However, those mRNAs that interacted with the microtubule were just as likely to undergo long-distance transport as those incubated without Staufen (Fig. 2 D). The presence of Staufen also reduced the number of processive dynein complexes and the proportion of microtubule-associated dyneins that underwent processive movement (Fig. 2, E–G). We conclude from these experiments that Staufen directly inhibits transport of *oskar* by Egl, BicD, dynactin, and dynein and that this effect is associated with impaired activation of motor movement.

Dynein movement during mRNA transport is switched on by the N-terminal coiled-coil region of BicD, which recruits dynactin (and thereby its activating functions) to the motor complex (Dienstbier et al., 2009; McKenney et al., 2014; Schlager et al., 2014; Hoogenraad and Akhmanova, 2016). These events are triggered by docking of RNA-bound Egl to the C-terminal region of BicD, which frees the N-terminal region from an autoinhibitory interaction (McClintock et al., 2018; Sladewski et al., 2018). To better resolve how dynein activity is impaired by Staufen, we assessed its effect on dynein activation by the N-terminal coiled-coil of the mouse BicD orthologue BicD2 (BicD2N), which is constitutively active in the absence of cargo (Hoogenraad et al., 2003; Dienstbier et al., 2009; McKenney et al., 2014; Schlager et al., 2014; Hoogenraad and Akhmanova, 2016). In the presence of dynactin and BicD2N, Staufen partially reduced the number of processive dynein movements and the

proportion of processive dyneins (Fig. 2, H and I; and Fig. S2 D). However, Staufen's inhibitory effect on the motor was proportionally much stronger in the presence of *oskar*, Egl, full-length BicD, and dynactin (Fig. 2, H and I; and Fig. S2 D). These data suggest that Staufen inhibits both the activation of dynein-dynactin motility by BicD proteins, as well as stimulation of this event by Egl and RNA.

## A balance of Staufen and Egl is required for *oskar* mRNA localization in vivo

We next explored the interplay between Egl and Staufen function in vivo by manipulating the dosage of these proteins and determining the effects on *oskar* mRNA localization in mid-oogenesis. We evaluated mRNA distributions qualitatively (Fig. 3) and confirmed our conclusions through an established, unbiased statistical averaging method (Ghosh et al., 2012; Gaspar et al., 2014; Fig. S3). Similar to the situation in *stau* mutant egg chambers (Fig. 3, A and B; and Fig. S1 B; and Fig. S3, A and B), UAS/Gal4-mediated overexpression of Egl in the germline resulted in a strong *oskar* signal at the anterior of the oocyte cortex, and a weaker enrichment of the mRNA at the posterior (Fig. 3 C and Fig. S3 C). This finding corroborates recent observations in Egl-overexpressing egg chambers by Mohr et al. (2021).

Altogether, these observations suggest that Staufen and Egl antagonize each other during *oskar* mRNA localization in mid-oogenesis. To further test this idea, we overexpressed both GFP-Staufen (using the maternal tubulin promoter [Micklem et al., 2000]) and Egl in the germline and examined *oskar* localization. While GFP-Staufen overexpression in this manner had no major effect on the localization of *oskar* (Fig. S1, B, C, and E; and Heber et al., 2019), it suppressed the ectopic anterior localization of *oskar* mRNA caused by elevated Egl levels (Fig. 3, C and D; and Fig. S3, C and D). This observation is consistent with mutual antagonism between Egl and Staufen. Further supporting this notion, removal of a functional copy of *egl* partially suppressed *oskar* mislocalization at the anterior of egg chambers with reduced Staufen levels (*stau* RNAi, *egl*[l]/+; Fig. S3, E–G). Furthermore, overexpression of Staufen suppressed the ectopic anterior localization of *oskar* mRNA caused by two different dominant BicD alleles (Fig. 3, E and F; and Fig. S3, H–K). At least one of these mutations (BicD[1]) appears to suppress the autoinhibited state of BicD in which the dynein and Egl binding sites are folded back on one another (Liu et al., 2013). Antagonism between BicD and Staufen is further supported by the observation that reduced *stau* gene dosage enhances embryonic lethality associated with the dominant BicD alleles (Navarro et al., 2004). Collectively, these genetic interaction experiments indicate a mutually antagonistic relationship between Staufen and Egl, as well as the latter protein's binding partner, BicD, during the localization of *oskar* mRNA in mid-oogenesis.

## Staufen dissociates Egl from *oskar* RNPs in the oocyte

To shed light on how Staufen and Egl influence each other in vivo, we examined the association of each protein with *oskar* mRNA using smFISH in combination with transgenically expressed, fluorescently tagged proteins (Gáspár et al., 2017; Heber et al., 2019).

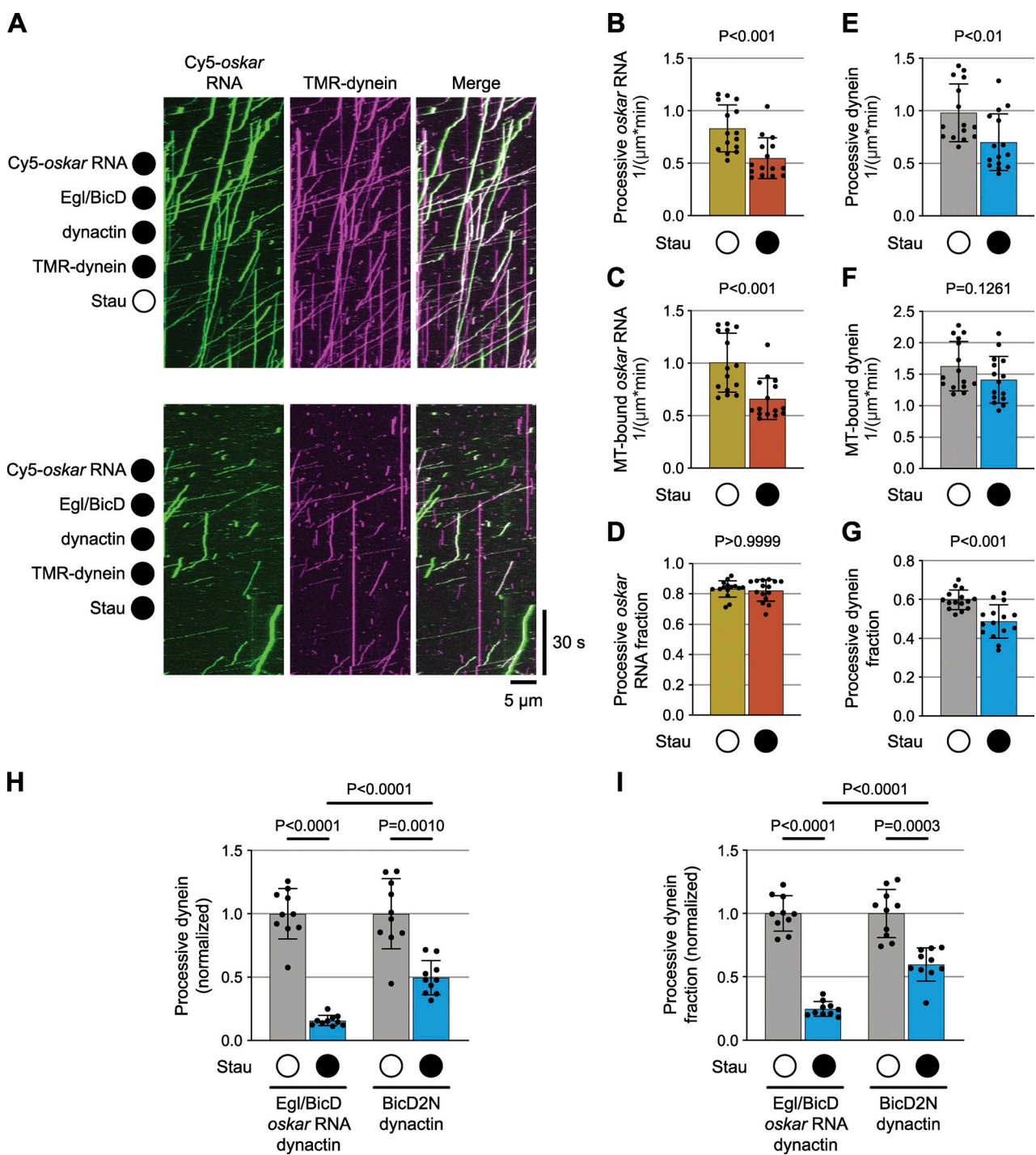

Figure 2. **Staufen impairs *oskar* mRNA transport and dynein activation in vitro. (A)** Kymographs (time-distance plots) exemplifying the behavior of *oskar* RNA and dynein in the presence (filled circle) and absence (open circle) of Staufen; in both conditions, Egl, BicD, and dynactin are also present but not fluorescently labeled. Minus end is to the left and plus end to the right. **(B–G)** Quantification of motile properties of *oskar m*RNA (B–D) and dynein (E–G) in the conditions shown in A. Charts show frequency of processive movements (B and E), total number of microtubule (MT) binding events (C and F), and fraction of microtubule-binding events that result in processive motility (D and G). Plots show the mean ± SD of values from 15 individual microtubules (represented by black circles) derived from analysis of 586–1,341 single RNA particles (B–D) or 1,247–2,207 single dynein particles (E–G) per condition. Statistical significance and P-values were determined with Mann–Whitney tests. **(H–I)** Quantification of effect of Staufen on number of processive dynein complexes and fraction of dynein-microtubule binding events that result in processive motility for motor activated by dynactin, *oskar* RNA, and Egl/BicD (H) vs. motor activated by dynactin and BicD2N (I). Values for conditions with Staufen were normalized to the corresponding condition without Staufen to obtain relative metrics. Plots show the mean ± SD of values from 10 individual microtubules (represented by black circles) derived from analysis of 536-925 single dynein particles per condition. Statistical significance and P-values were determined with Brown-Forsythe and Welch ANOVA tests.

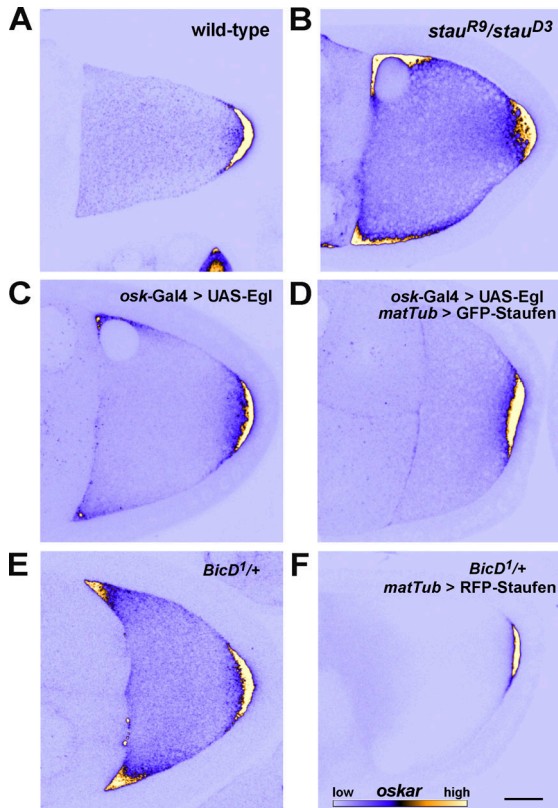

**Figure 3. Mutual interference of Staufen and Egl during *oskar* mRNA localization. (A–F)** Representative micrographs of the distribution of endogenous *oskar* mRNA (detected by smFISH; blue—low intensity RNA signal, yellow—high intensity RNA signal, see scale in F) in stage 9 oocytes (anterior to the left, posterior to the right) showing ectopic anterior accumulation of *oskar* mRNA in oocytes lacking Staufen protein (B; see also Fig. 1 A′), over-expressing Egl (C) or heterozygous for one copy of the dominant, hyperactive *BicD1* allele (E). Anterior accumulation of *oskar* is not observed in wild-type oocytes (A), or upon Staufen overexpression in Egl-overexpressing oocytes (D) or *BicD1/+* oocytes (F). Scale bar represents 20 µm.

Distribution of transgenic Staufen-GFP, expressed by the *stau* promoter at close to endogenous levels (Fig. S1 F), resembles that of the endogenous protein (St Johnston et al. 1991), and this molecule is functional during *oskar* localization (Fig. S1, B and C). In the oocyte, Staufen-GFP and *oskar* mRNA frequently overlapped. In the nurse cells, there was little overlap between Staufen-GFP and *oskar*, as previously reported (Little et al., 2015), and only low levels of Staufen-GFP signal were observed, regardless of the developmental stage of the egg chamber (Fig. 4, A, A′, and B). In the oocyte, the fraction of Staufen-associated *oskar* RNPs increased as a function of their *oskar* mRNA content (which was determined by calibration of smFISH signals [see Materials and methods and Little et al., 2015]; Fig. S4 A). This increase was substantially more pronounced during stage 9, when there is a strong enrichment of *oskar* at the oocyte posterior, than in earlier stages of oogenesis (Fig. 4 B). During stage 9, not only the likelihood of Staufen association with *oskar* RNPs increased, but also the relative level of Staufen per RNP (Fig. 4 B′ and Fig. S4 B). We previously observed the same behavior for overexpressed GFP-Staufen and mammalian GFP-

Stau2 (Heber et al., 2019). Notably, however, no or greatly reduced scaling of the Staufen signal with *oskar* mRNA content of the RNPs was observed in the oocyte prior to the onset of *oskar* localization to the posterior (Fig. 4 B′). This is likely due to limited availability of the protein, as suggested by the relatively low fluorescent signal in early stage Staufen-GFP oocytes (Fig. S4 C). Collectively, our analysis shows that Staufen preferentially associates with *oskar* mRNA in the oocyte and that this association is largely governed by the RNA content of the RNPs and the enrichment of the pool of Staufen protein in the ooplasm.

During the same stages of oogenesis, Egl-GFP (which was mildly overexpressed with the α-tubulin84B promoter [Fig. S4 D; Dienstbier et al., 2009]) was abundant in both the nurse cells and the oocyte (Fig. 4, C and C′). This pattern matches the distribution of endogenous Egl protein, as revealed by immunostaining (Mach and Lehmann, 1997). Colocalization analysis showed that, in contrast to Staufen-GFP, Egl-GFP was already associated with *oskar* RNPs in the nurse cells (Fig. 4 D) and maintained its association with *oskar* in the oocyte until the onset of posterior localization. However, from stage 9 onward, we detected no significant colocalization between Egl and RNPs containing four or more copies of *oskar* mRNA (Fig. 4 D). The amount of Egl signal associated with the RNPs increased proportionately with *oskar* mRNA content during early oogenesis but reduced by about twofold and became independent of *oskar* mRNA copy number during stage 9 in the oocyte (Fig. S4 E). As the loss of Egl from *oskar* RNPs was an unexpected finding, we wondered if it was caused by the dilution of Egl-GFP by endogenous, untagged Egl. However, this was not the case as the reduction of Egl association with *oskar* RNPs was robust to changes in the dosage of the Egl-GFP transgene or endogenous unlabeled Egl, and was also observed when no unlabeled Egl (*egl1/egl2* rescued by the Egl-GFP transgene) was present in the oocyte (Fig. S4, F and F′). Collectively, these results indicate that, whereas Staufen levels increase on *oskar* RNPs as posterior-ward movement in the oocyte is initiated, there is a concomitant decrease in Egl levels on these complexes.

The opposing trends of Staufen and Egl association with *oskar* RNPs in the nurse cells and the oocyte (Fig. 4, B, B′, D, and D′), together with the increasing concentrations of Staufen in the ooplasm prior to the onset of *oskar* localization (Fig. S4 C), are consistent with Staufen inhibiting Egl association with *oskar* RNPs in the oocyte. To test this notion, we analyzed the association of Egl-GFP with *oskar* RNPs when Staufen was disrupted. When Staufen protein was absent or greatly reduced, Egl remained associated with *oskar* RNPs in stage 9 oocytes, and the relative amount of Egl signal that colocalized with the RNPs increased with *oskar* mRNA content (Fig. 4, E and E′; and Fig. S4, G and G′). Conversely, when Staufen was mildly overexpressed using an RFP-Staufen transgene (Zimyanin et al., 2008), the amount of Egl associated with *oskar* RNPs in the oocyte (in particular with those containing 4+ copies of RNA) decreased (Fig. 4, F and F′).

To test biochemically whether Staufen impairs the association of Egl with *oskar* RNPs in vivo, we performed UV crosslinking and GFP immunoprecipitation followed by quantitative

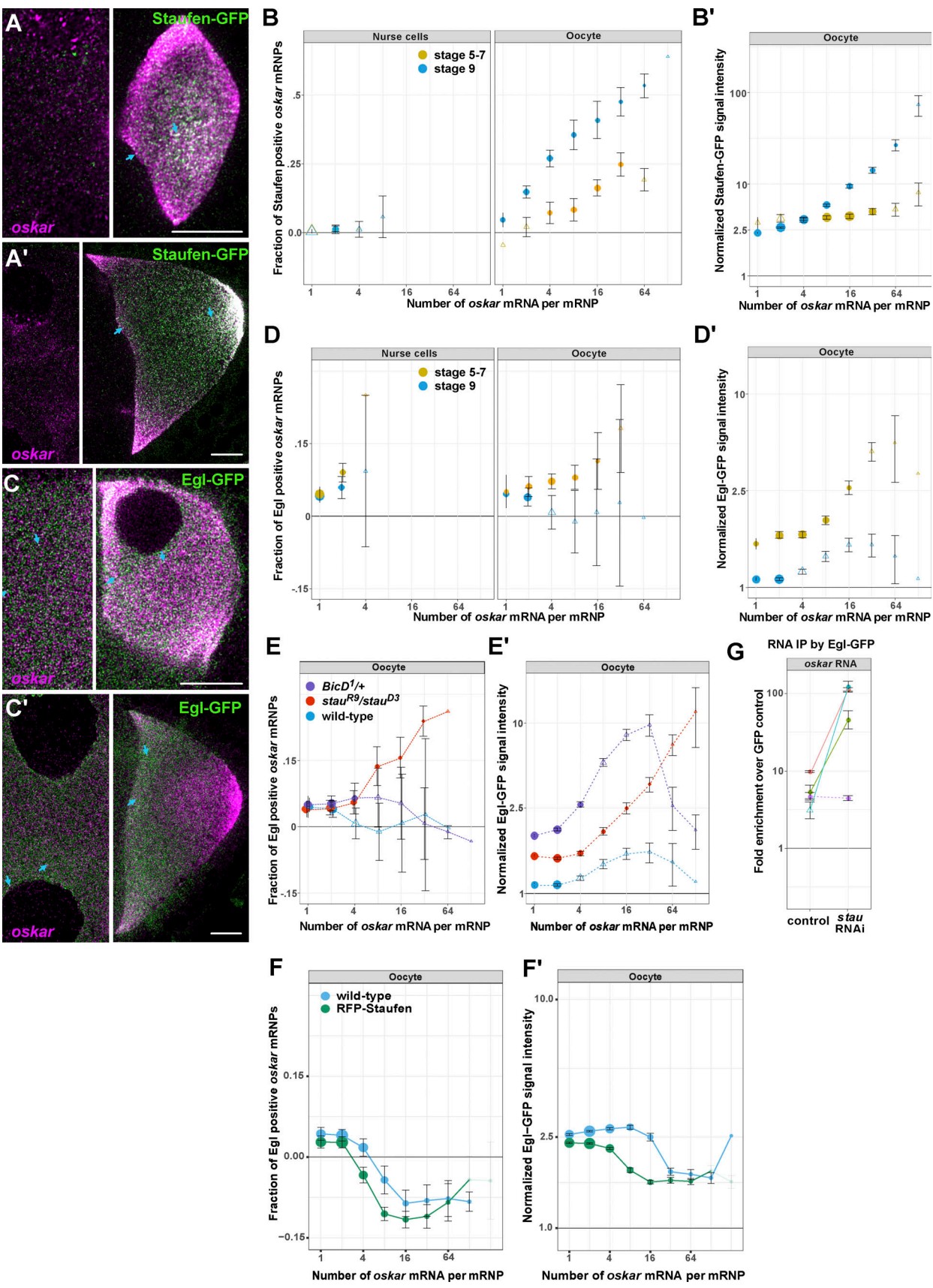

Figure 4. **Staufen interferes with association of Egl with *oskar* RNPs in the oocyte. (A and A')** Staufen-GFP (green) distribution in the nurse cell cytoplasm (left-hand images) and oocyte (right-hand images) before (stage 6–7, A) and during (stage 8–9, A') the posterior localization of endogenous *oskar*

mRNA (magenta) in the oocyte of the same egg chamber. **(B and B')** Quantification of Staufen-GFP association with *oskar* RNPs in nurse cell cytoplasm and ooplasm at the indicated stages as a function of RNA copy number. The number of analyzed *oskar* RNPs was 3,105 (stage 5–7 nurse cells), 8,749 (stage 9 nurse cells), 3,814 (stage 5–7 oocytes) and 14,054 (stage 9 oocytes). **(C and C')** Distribution of Egl-GFP (green) in the nurse cell cytoplasm (left-hand images) and oocyte (right-hand images) before (stage 7, C) and during (stage 9, C') the posterior localization of endogenous *oskar* mRNA (magenta) in the oocyte of the same egg chamber. In A, A', C, and C', nurse cell images are shown with different brightness/contrast settings for better visualization of the signals. Cyan arrows point to examples of colocalization. Scale bars represent 10 μm. **(D and D')** Quantification of association of Egl-GFP with *oskar* RNPs in nurse cell cytoplasm and ooplasm at the indicated stages. The number of analyzed *oskar* RNPs was 8,511 (stage 5–7 nurse cells), 10,210 (stage 9 nurse cells), 6,897 (stage 5–7 oocytes), and 15,869 (stage 9 oocytes). **(E–F')** Association of Egl-GFP with *oskar* RNPs in the stage 9 oocyte in the indicated genotypes. In A–B', endogenous Staufen was absent (*stau^R9^/stau^D3^* background), whereas in C–F' endogenous, unlabeled Egl was present. 15,869–26,232 *oskar* RNPs were analyzed for each genotype. Error bars represent 95% confidence intervals in B, B', and D–F'. In D and E, datapoints are slightly offset in the x-axis to facilitate comparison. The size of the circles is proportional to the relative abundance of each category of *oskar* RNPs within the overall population. Triangles indicate that the fraction of GFP-positive *oskar* RNPs is not significantly different from zero (P > 0.01, one sample *t* test, B, B', and D–F'). In B', D', E', and F', GFP protein intensities on *oskar* RNPs in the oocytes are normalized to GFP signal intensities in the corresponding nurse cells. **(G)** Fold-enrichment of *oskar* mRNA precipitated from UV crosslinked Egl-GFP ovarian extracts relative to the GFP control in control and *stau* RNAi. Different colors represent different experiments (four in total). Solid (P < 0.05) or dashed lines (P > 0.05) connect paired data (pairwise Student's *t* test was used to determine significance). Empty triangles indicate non-significant enrichment (P > 0.05) of *oskar* relative to the GFP control. Error bars show SD of three replicates per experiment.

RT-PCR on egg chambers co-expressing Egl-GFP and either *stau* RNAi or a control RNAi in the germline. Substantially more *oskar* mRNA was co-immunoprecipitated with Egl-GFP from extracts of egg chambers expressing *stau* RNAi compared to the control (Fig. 4 G and Fig. S4 H). We conclude from these experiments that Staufen antagonizes the association of Egl with *oskar* mRNA in vivo.

### Aberrant anterior localization of *oskar* mRNA correlates with increased Egl association

As described above, BicD is a direct binding partner of Egl, and contacts the dynein and dynactin complexes (Dienstbier et al., 2009; Liu et al., 2013; Vazquez-Pianzola et al., 2017; McClintock et al., 2018; Sladewski et al., 2018). To gain further insight into how the association of Egl influences trafficking of *oskar* mRNA, we examined the effect of the two gain-of-function *BicD* mutations in more detail. Both alleles increased the association of Egl-GFP with mid- to large-sized (4–16 RNA copies) *oskar* RNPs in stage 9 oocytes when compared to the wild-type control (Fig. 4, E and E'; and Fig. S4, F and F'). Although the magnitude of the effect differed between the two *BicD* alleles, in both cases the amount of Egl signal on the RNPs scaled with *oskar* mRNA copy number up to 32 copies, in contrast with the low amount of Egl detected on *oskar* RNPs in wild-type stage 9 egg chambers (Fig. 4 E' and Fig. S4 F'). Thus, there is a correlation between the ability of the BicD gain-of-function alleles to misdirect *oskar* RNPs to the anterior of the oocyte and their stimulation of Egl association with these structures in the oocyte cytoplasm.

We reasoned that the anterior localization of *oskar* mRNA in the oocyte observed when untagged Egl was strongly overexpressed (Fig. 3 C and Fig. S3 C) might be due to an aberrant, concentration-dependent retention of the protein on the RNPs. To test this idea, we introduced the moderately overexpressed Egl-GFP transgene (Dienstbier et al., 2009) into egg chambers overexpressing untagged Egl under UAS-Gal4 control (Fig. S4 D). We observed numerous large RNP granules containing high amounts of Egl-GFP (which presumably also contain unlabeled Egl) and *oskar* mRNA (Fig. 5, A and A') in the nurse cells and at the anterior cortex of the oocyte. Such RNPs were seldom detected in the absence of UAS/Gal4-mediated Egl overexpression (Fig. 5 E). Quantitative analysis revealed that the relative Egl-GFP signal scaled with *oskar* mRNA content in the

RNPs when untagged Egl was overexpressed (Fig. 5 C'). The effect was similar to that observed in the absence of Staufen, as well as in egg chambers with *BicD* gain-of-function alleles (Fig. 4 E' and Fig. S4 F'). In contrast, we found no significant change in Staufen association with *oskar* RNPs in stage 9 oocytes strongly overexpressing Egl (Fig. 5, B, B', D, and D'). This finding indicates that Egl does not overtly interfere with the loading of Staufen on *oskar* RNPs. However, substantially more high copy number *oskar* RNPs (containing 17+ copies of *oskar*) were localized at the anterior pole when Egl was strongly overexpressed than in the wild type (Fig. 5 E and Fig. S5), consistent with our earlier finding that Egl antagonizes Staufen's function in promoting posterior localization of *oskar* mRNA.

Taken together, these data reveal that a shared feature of *oskar* mislocalization to the oocyte anterior in the absence of Staufen, upon Egl overexpression, and in gain-of-function *BicD* mutants is the failure to release Egl from the RNPs during mid-oogenesis. Our data indicate that this release is mediated by Staufen, which interferes with the association of Egl with *oskar* RNPs, especially those with high *oskar* mRNA content.

### Dissociation of Egl does not prevent association of dynein with *oskar* RNPs

As described above, Egl functions as an adaptor molecule linking mRNA localization signals to dynein-dynactin (Navarro et al., 2004; Dienstbier et al., 2009; Amrute-Nayak and Bullock, 2012). However, it has been proposed that Egl is not the only means to link the motor complex to RNPs. In ex vivo experiments, both the *K10* and *hairy* mRNAs recruit some dynein-dynactin complexes through Egl-BicD, and others through an Egl-BicD independent mechanism, which presumably involve other RBP linkers or direct interactions between the RNA and the motor complex (Bullock et al., 2006; Amrute-Nayak and Bullock, 2012; Dix et al., 2013; Soundararajan and Bullock, 2014). Only those dynein-dynactin complexes interacting with Egl-BicD were capable of long-distance transport, explaining how these two adaptors promote minus-end-directed trafficking of mRNAs. How Egl influences dynein's association with RNPs has not, however, been examined in vivo.

To address this question, we quantified the colocalization of *oskar* mRNA (detected by smFISH) with fluorescently tagged

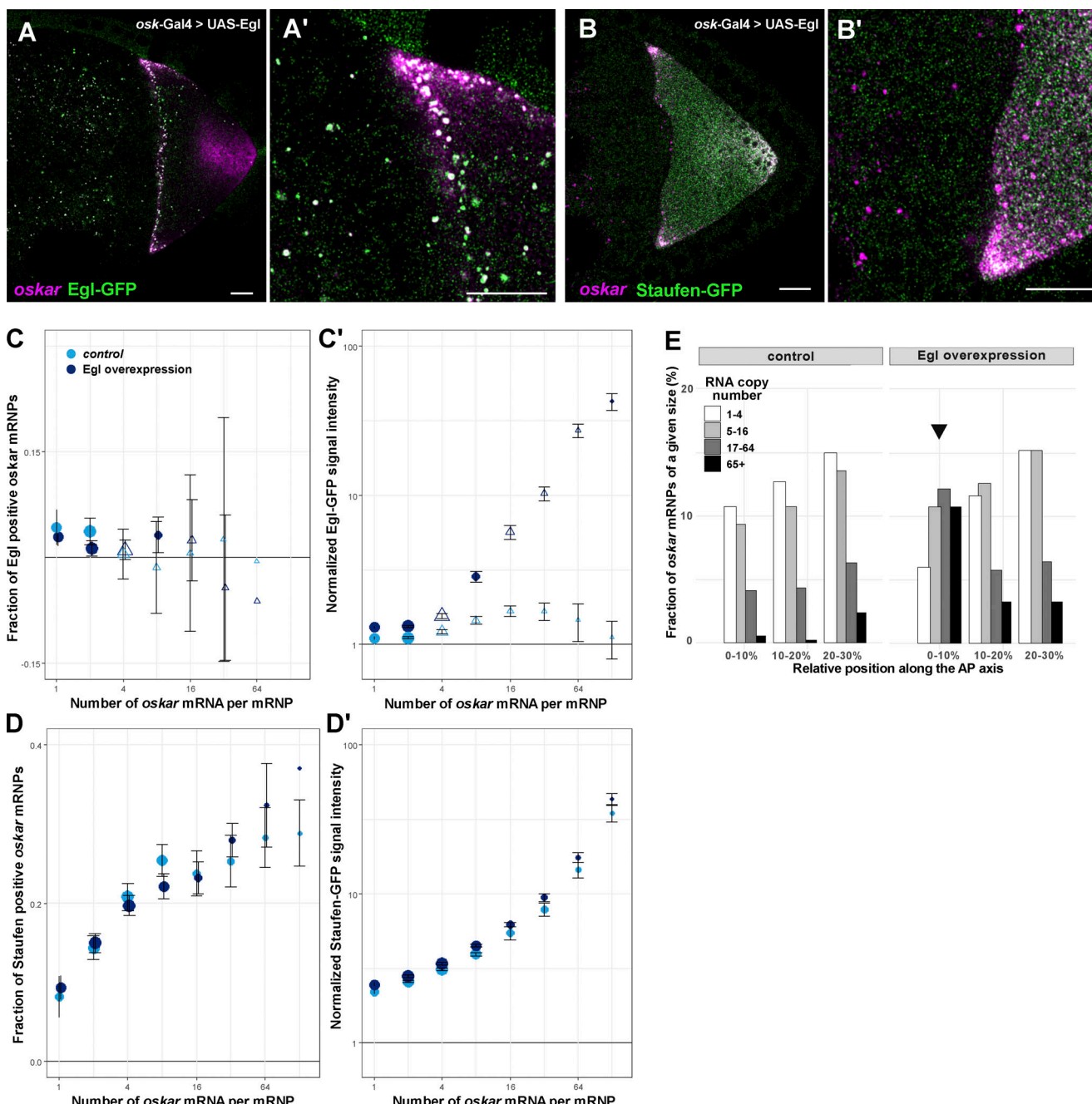

Figure 5. **Effect of Egl overexpression on *oskar* RNP localization and composition. (A–B')** Images of *oskar* mRNA with Egl-GFP (A and A') or Staufen-GFP (B and B') in stage 9 egg chambers strongly overexpressing Egl in the germline (*osk-Gal4>UAS-Egl*). In both genotypes, abnormally large RNPs containing *oskar* mRNA are observed in the nurse cells and in the anterior region of the oocyte. These RNPs frequently colocalize with Egl-GFP but rarely co-localize with Staufen-GFP. Scale bar represents 10 μm. **(C–D')** Quantification of Egl-GFP and Staufen-GFP association with *oskar* RNPs as a function of RNA copy number. In C and D, datapoints are slightly offset in the x-axis to facilitate comparison. Error bars represent 95% confidence intervals and the size of the circles is proportional to the relative abundance of each category of *oskar* RNP within the overall population. Triangles indicate that the fraction of GFP-positive *oskar* RNPs is not significantly different from zero (P > 0.01, one sample *t* test, C–D'). 26,432–33,698 *oskar* RNPs per genotype were analyzed. **(E)** Relative distribution of *oskar* RNPs grouped by RNA content along the first 30% of the anteroposterior axis of stage 9 oocytes. In the wild-type control, <1% of large (65+ copies) *oskar* RNPs are close to the oocyte anterior (first bin). When Egl is overexpressed, ∼10% of large *oskar* RNPs are near the oocyte anterior (first bin, arrow), while the rest of the oocyte has a distribution of *oskar* RNPs similar to the control (Fig. S5). 27,946 and 33,698 *oskar* RNPs in control and Egl overexpressing oocytes were analyzed, respectively.

versions of BicD (Fig. 6 A), the dynactin subunit p50/dynamitin a dynactin subunit (Fig. 6 B), and Dynein heavy chain (Dhc; Fig. 6 C) in the nurse cells and oocyte. We saw little difference in association of each protein with *oskar* mRNA between the stage 9 nurse cells and oocyte (Fig. 6 D). This contrasts with the situation for Egl, which showed a relative reduction in association with *oskar* in the stage 9 oocyte (Fig. 4, D and D'). The fraction of *oskar* RNPs associated with BicD and the dynein-dynactin subunits also did not change as a function of RNA content in the oocyte (Fig. 6 D). Notably, the relative amount of these proteins associated with *oskar* RNPs in the oocyte tended to be approximately double that in the nurse cells (Fig. 6 E). The amount of BicD associated with the RNPs in the oocyte was independent of *oskar* copy number. In contrast, p50 and Dhc association increased as a function of mRNA content, although not proportionally (~two to fivefold increase in signal from the respective proteins versus an ~100-fold increase in *oskar* mRNA content; Fig. 6 E). This result indicates that BicD, p50, and Dhc are not recruited to *oskar* RNPs by identical mechanisms.

Increased levels of Egl on *oskar* RNPs when Staufen is disrupted (Fig. 4, E and E'; and Fig. S4, G and G') or when Egl is overexpressed (Fig. 5, C and C') could in principle result in increased recruitment of the dynein motor. However, when we analyzed GFP-Dhc association with *oskar* RNPs, we found no major difference between *stau* RNAi, Egl-overexpressing, and control oocytes (Fig. 6, F and F'). These data indicate that Egl is not obligatory for recruitment of dynein-dynactin to a native mRNA cargo in vivo. Intriguingly, *stau* RNAi oocytes also did not display an increase in the association of BicD with *oskar* RNPs (Fig. 6, G and G'). Thus, there appears to be an Egl-independent mechanism for recruiting BicD to these structures.

The loss of Egl from larger *oskar* RNPs during stage 9 suggests that dynein—although present—is not fully active. This notion is consistent with our earlier observation in control ooplasmic extracts that RNPs with a greater RNA content, and which therefore have less Egl bound (Fig. 1 B; and Fig. 4, D and D'), engaged in minus-end-directed motion less frequently than those with a lower RNA content: there was an approximately fourfold plus end dominance (77.9 vs. 22.1% of plus- vs. minus-end-directed runs) for *oskar* RNPs with more than two relative RNA units, versus an approximately twofold plus end dominance (~65 vs. ~35%) for smaller RNPs (Fig. 1 B). In summary, in stage 9 oocytes, *oskar* RNPs lacking Egl maintain their association with the dynein machinery but are defective in minus-end-directed motion.

## Staufen activity is controlled by Egl-mediated delivery of its mRNA into the oocyte

As shown above, proper *oskar* localization to the posterior pole depends on availability of Staufen protein in stage 9 oocytes. However, it is unclear how this spatio-temporal restriction of Staufen is brought about. We noticed that in *egl* RNAi egg chambers only trace amounts of endogenous Staufen (Fig. 7, C, C', and D) or Staufen-GFP (Fig. S6 A) were present in the early arrested oocyte primordium, which was marked by residual accumulation of *oskar* mRNA (Fig. 7, A and A'). This observation

raised the possibility that Egl mediates the enrichment of Staufen in the ooplasm of wild-type egg chambers. The normal ooplasmic accumulation of Staufen could in principle result from its transport in association with target mRNAs from the nurse cells into the oocyte. This seems unlikely, however, as we detected no association of Staufen-GFP with its main target, *oskar* mRNA, in the nurse cells (Fig. 4, A–B'), and both Staufen and Staufen-GFP signal accumulated in the oocyte even in the absence of *oskar* mRNA (Fig. S6, A–C'). Alternatively, *stau* mRNA could be transported by Egl-BicD-dynein-dynactin into the oocyte, leading to accumulation of Staufen protein at this site. Supporting this notion, whereas *stau* mRNA and protein were strongly enriched in oocytes of early stage egg chambers expressing control RNAi, we observed little or no such enrichment in egg chambers in which *egl* was knocked down by RNAi (Fig. 7, B–D and Fig. S6 D). Furthermore, while *stau* mRNA did not display any specific localization pattern in the ooplasm of wild-type oocytes at mid-oogenesis (Fig. 7 E), upon Egl overexpression the transcript was enriched at the anterior pole of the oocyte (Fig. 7 E'). These observations suggest that, similar to other oocyte-enriched transcripts, such as *oskar, gurken,* and *bicoid* (Ephrussi et al., 1991; Berleth et al., 1988; Neuman-Silberberg and Schüpbach, 1993; Theurkauf et al., 1993; Clark et al. 2007), *stau* mRNA is a target of Egl-dependent transport from the nurse cells by dynein.

Providing further support for this notion, colocalization analysis revealed that *stau* mRNA and Egl-GFP were associated in the germline, and that this association was significantly lower in the oocyte than in the nurse cells (Fig. 7 F). We also detected Staufen protein in association with *stau* RNPs in stage 9 oocytes (Fig. 7 G), as we previously observed for *oskar* RNPs (Fig. 4 B).

Additional experiments provided evidence that the restriction of Staufen activity to the oocyte is critical for proper RNA localization and oogenesis. In contrast to the direct expression of GFP-Staufen from the maternal tubulin promoter, which results in nurse cell accumulation of the protein only from mid-oogenesis onwards (Fig. S6 E), strong overexpression of GFP-Staufen with highly active maternal Gal4 drivers results in high levels and aggregation of the protein in the nurse cells of early egg chambers (Fig. S6, G–I'). Ectopic GFP-Staufen expression in this manner resulted in failure of *oskar*, as well as *bicoid*, mRNA enrichment in the oocyte (Fig. S6, F–G″). These egg chambers also displayed oocyte polarity defects and fragmentation of the karyosome (Fig. S6, H and I), which are hallmarks of mislocalization of dynein's mRNA cargoes or loss of dynein activity (Jenny et al. 2006; Neuman-Silberberg and Schüpbach 1993; Januschke et al. 2002).

Taken together, these data indicate that the Egl-dependent transport of *stau* mRNA into the oocyte by dynein underlies ooplasmic enrichment of Staufen protein, which in turn inactivates the machinery for minus-end-directed transport of *oskar* RNPs in the stage 9 oocyte. This constitutes a feed-forward mode of regulation, whereby the dynein transport system ensures the localized production of its inhibitor, leading to its own inactivation. This predefined program thereby allows kinesin-1-dependent delivery of *oskar* mRNA to the oocyte posterior.

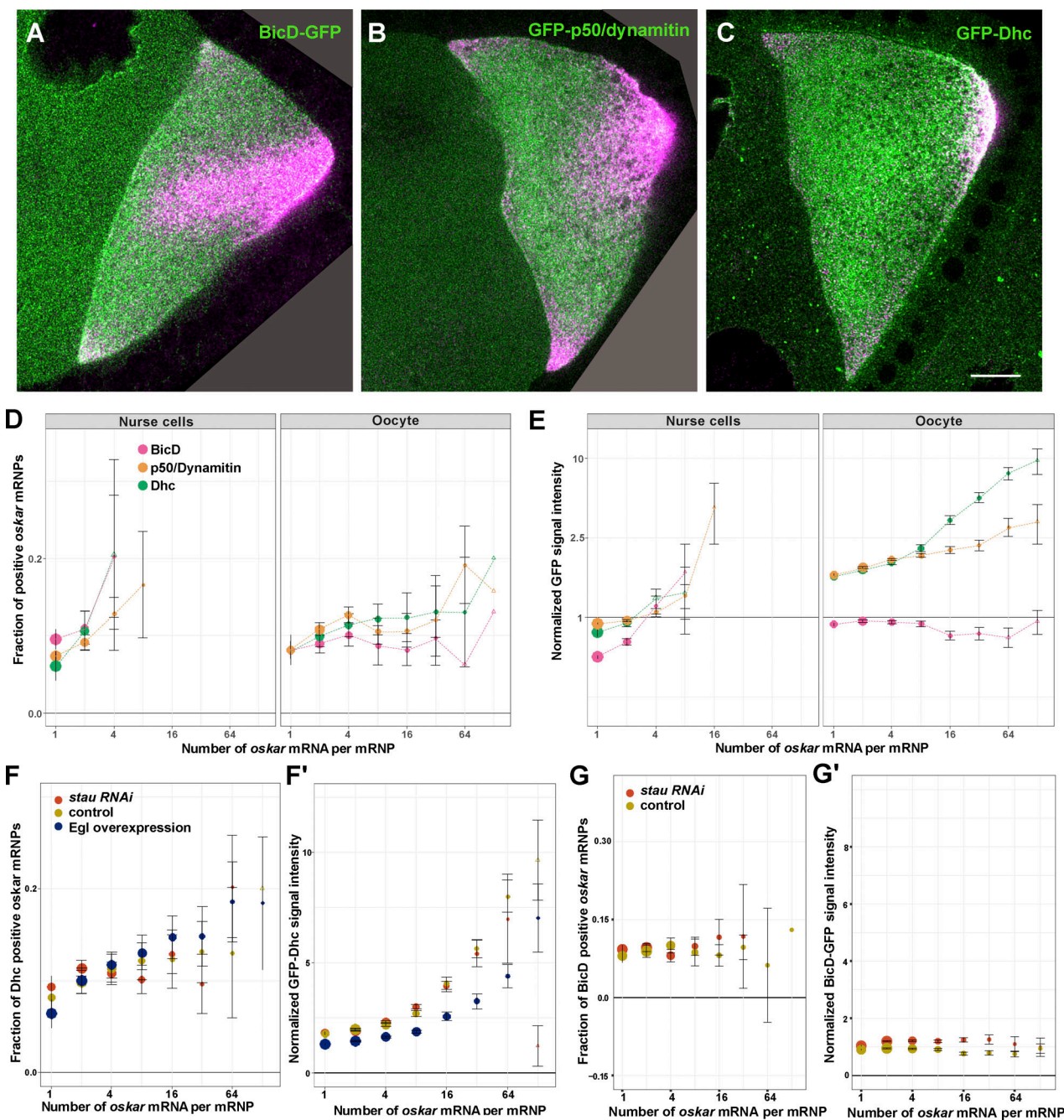

Figure 6. **Association of the dynein machinery with *oskar* RNPs. (A–C)** Localization of GFP tagged versions (green) of BicD (A), p50/dynamitin (B), and Dhc (C) with respect to *oskar* mRNA (magenta) in egg chambers during mid-oogenesis. Scale bar represents 10 µm. Note that the transient *oskar* accumulation in the center and at the anterior pole of early stage 9 oocytes is normal and disappears by the end of stage 9. Empty regions created upon rotating the images are shown in dark gray (A and B). **(D and E)** Quantification of *oskar* RNP association with the indicated proteins in the nurse cells and the oocyte as a function of RNA copy number. The number of analyzed *oskar* RNPs was 12,945 (GFP-Dhc, nurse cells), 34,428 (GFP-p50/dynamitin, nurse cells), 8,573 (BicD-GFP, nurse cells), 27,936 (GFP-Dhc, oocytes), 37,992 (GFP-p50/dynamitin, oocytes), and 20,456 (BicD-GFP, oocytes). **(F–G′)** Quantification of *oskar* RNP association with GFP-Dhc and BicD-GFP in oocytes of indicated genotypes. 29,428, 32,132 and 33,638 *oskar* RNPs were analyzed in control, *stau* RNAi and Egl overexpressing oocytes, respectively (F′). 20,456 and 32,517 *oskar* RNPs were analyzed in control and *stau* RNAi oocytes, respectively (G′). Error bars represent 95% confidence intervals, and the size of the circles is proportional to the relative abundance of each category of *oskar* RNP within the overall population. In D, F, and F′, datapoints are slightly offset in the x-axis to facilitate comparison. Triangles indicate that the fraction of GFP-positive *oskar* RNPs is not significantly different from zero (P > 0.01, one sample *t* test, D–G).

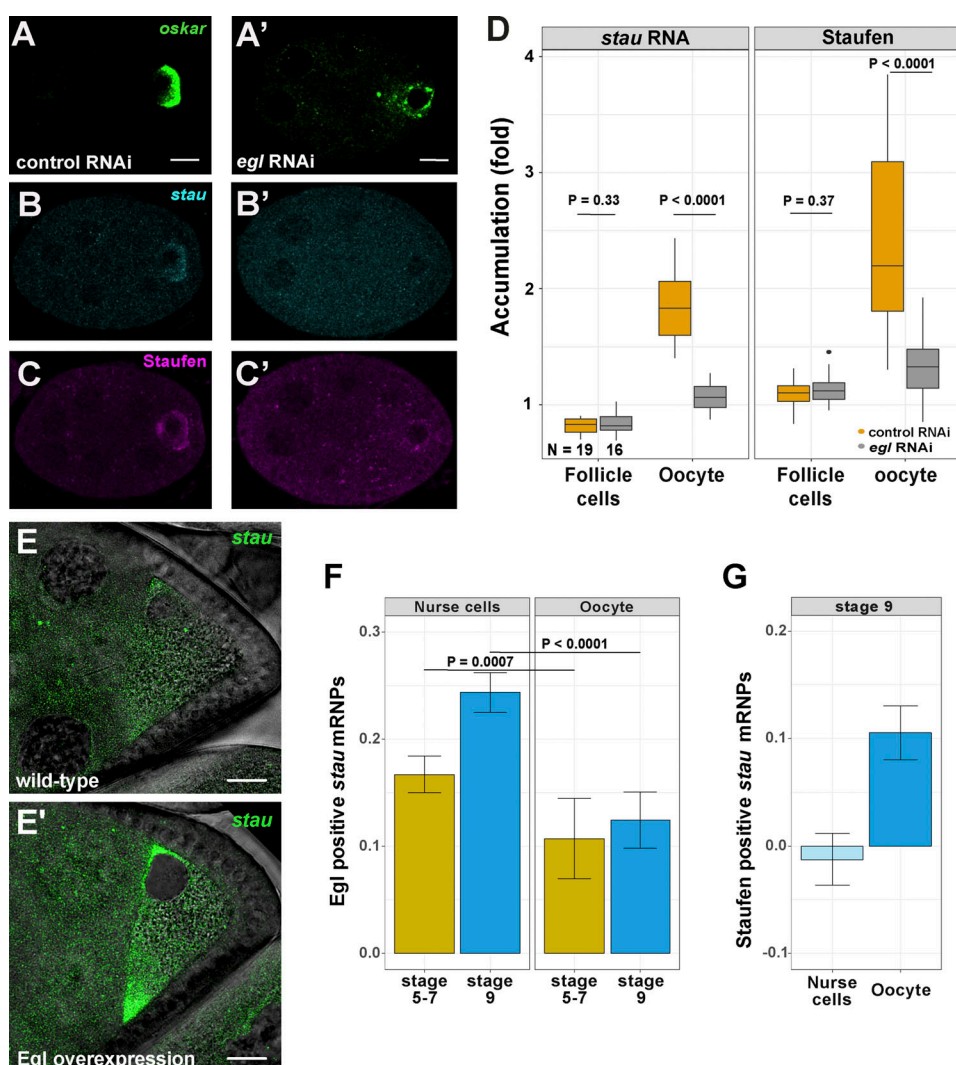

Figure 7.  **Egl promotes ooplasmic enrichment of Staufen mRNA and protein. (A–C')** Distribution of endogenous Staufen protein (A and A'; magenta), *oskar* mRNA (B and B'; green), and *stau* mRNA (C and C'; cyan) in early egg chambers (stage 4–6) expressing control RNAi (A–C) or *egl* RNAi driven by *osk*-Gal4 (A', B', and C'). Oocytes of *egl* RNAi egg chambers contain trace amounts of *oskar* mRNA, likely due to the action of residual Egl protein. **(D)** Quantification of early oocyte enrichment of *stau* mRNA or Staufen protein relative to the sibling nurse cells. Stage 4–6 oocytes were identified through their enrichment of *oskar* mRNA. Enrichment of *stau* RNA or Staufen protein in the somatic follicle cells, which do not express the shRNA, is used as a control. 19 control RNAi and 16 *egl* RNAi samples were analyzed, respectively (also indicated on the panel). **(E and E')** Localization of *stau* mRNA in wild-type and Egl-overexpressing stage 8 oocytes. **(F and G)** Fraction of *stau* RNPs associating with Egl in the nurse cells and in the oocyte (F), and Staufen in the oocyte (G). **(F)** The number of analyzed *stau* RNPs was 4,075 (stage 5–7 nurse cells), 3,420 (stage 9 nurse cells), 1,283 (stage 5–7 oocytes), and 1,455 (stage 9 oocytes). **(G)** 8,597 and 4,114 *stau* RNPs were analyzed in nurse cells and in oocytes, respectively. Transparent bar for nurse cells indicates non-significant difference to zero (P > 0.01, one sample *t* test; D and F). P values of unpaired, two-sample Student's *t* test are shown. Scale bars represent 10 μm.

## Discussion

The activity of motor molecules is essential for establishing proper intracellular localization of mRNA molecules, which in turn underlies spatiotemporal restriction of protein function (St Johnston, 2005; Martin and Ephrussi, 2009; Mofatteh and Bullock, 2017; Abouward and Schiavo, 2021). The dynein and kinesin microtubule motors play key roles in positioning of mRNAs in many systems, including by acting sequentially on the same RNP species (Baumann et al., 2012; Gagnon et al., 2013; Mofatteh and Bullock, 2017; Turner-Bridger et al., 2018). However, it is unclear how the opposing activities of these motors are coordinated during RNP trafficking.

Here we use the tractable *Drosophila* egg chamber to reveal a mechanism for spatiotemporal control of dynein and kinesin-1 activity. Central to this system are two dsRBPs, Staufen, and Egl. Genetic interaction experiments have previously shown that these proteins have opposing activities in the context of *oskar* mRNA localization in the oocyte and anteroposterior patterning (Mohler and Wieschaus, 1986; Navarro et al., 2004), and both proteins were shown to complex with *oskar* mRNA (Laver et al., 2013; Sanghavi et al., 2016). Recent work by Mohr and colleagues (Mohr et al., 2021), which was published while this manuscript was in preparation, further corroborated the ability of Egl to antagonize localization of *oskar* to the posterior of the

oocyte. Mohr et al. (2021) also showed that Egl binds in vitro to a double-stranded region in the *oskar* 3′ UTR that can enrich the mRNA at the anterior cortex of the oocyte, which they termed the Transport and Anchoring Signal (TAS). The TAS partially overlaps with one of the Staufen Recognized Structures (SRS; Laver et al., 2013) which are important for Staufen binding in vitro and for posterior *oskar* mRNA localization in the oocyte (Mohr et al., 2021). These data suggest that Staufen could antagonize Egl function by interfering with binding of the latter protein to *oskar* mRNA. However, Mohr et al. (2021) did not directly test this possibility. We found that knocking down Staufen increases the association of Egl with the mRNA, demonstrating the role of Staufen in antagonizing association of Egl with *oskar* mRNA. The results of our in vitro motility assays are also consistent with this scenario, as RNA binding to Egl is a prerequisite for full dynein activity within this minimal RNP (McClintock et al., 2018; Sladewski et al., 2018). It is conceivable that the transport and anchoring functions ascribed to the TAS are two facets of the same underlying molecular mechanism, as increased dynein activity in the absence of Staufen would drive enrichment of *oskar* RNPs at microtubule minus ends that are nucleated at the anterior cortex (Januschke et al., 2006).

Our study additionally reveals, through smFISH-based co-localization analysis in egg chambers, how the interactions of Staufen and Egl with *oskar* mRNA are orchestrated in time and space. When Staufen levels are low, such as in early oogenesis, amounts of Egl per RNP scale with *oskar* mRNA content. During stage 9, this scaling is lost in the oocyte and the relative amount of Egl on *oskar* RNPs decreases as Staufen is recruited to *oskar*. Our data indicate that Staufen-mediated displacement of Egl is a critical step in switching to kinesin-1-based trafficking of *oskar* mRNA to the posterior pole. This switch is also likely to involve the activities of exon junction complex components, which are required for posterior localization of *oskar* (Newmark and Boswell, 1994; Hachet and Ephrussi, 2001; Hachet and Ephrussi, 2004; Mohr et al., 2001; van Eeden et al., 2001; Palacios et al., 2004; Zimyanin et al., 2008; Ghosh et al., 2012). Our study provides a framework for understanding how the activities of these factors are coordinated with that of Staufen.

Intriguingly, whilst an excess of Staufen can dissociate Egl from *oskar* RNPs, increasing Egl concentration does not overtly affect Staufen recruitment to these structures. This observation might be due to the five-to-one excess of SRSs over the TAS in each *oskar* molecule (Mohr et al., 2021), which could mask the loss of Staufen binding to the TAS-proximal SRS.

We also found that dissociation of Egl from *oskar* RNPs in the stage 9 oocyte does not detectably alter the amount of dynein on these structures. This work lends in vivo support to the notion that additional proteins or RNA sequences can recruit dynein and dynactin in an inactive state to RNPs (Bullock et al., 2006; Amrute-Nayak and Bullock, 2012; Dix et al., 2013; Soundararajan and Bullock, 2014). Presumably, Egl recruits only a small fraction of the total number of dynein complexes on *oskar* RNPs or activates the motility of complexes that are linked to the RNA by other factors. Unexpectedly, we found that the bulk association of BicD with *oskar* RNPs is also not dependent on Egl, pointing to an additional mechanism for recruiting BicD, presumably in the

autoinhibited state (Hoogenraad et al., 2003; Liu et al., 2013). While the relative amount of the dynein machinery associated with *oskar* RNPs is greater in the oocyte than the nurse cells, association of the motor complex with *oskar* RNPs in the oocyte does not scale proportionally with RNA content. How the amount of dynein recruited to these RNPs could be limited is unclear. However, this mechanism could be a means to prevent sequestration of this multi-functional motor to a very abundant cargo (~0.5–1 million copies of *oskar* mRNA per oocyte [Little et al., 2015]).

Whilst our data build a strong case that a key function of Staufen in *oskar* mRNA localization is to limit dynein activity by displacing Egl, several lines of evidence suggest that this is not its only role in this context. We found that Staufen had a partial inhibitory effect on minus-end-directed motility of purified dynein-dynactin complexes activated in an Egl-independent manner by a constitutively active truncation of a BicD protein. Indeed, polarity defects observed in oocytes strongly over-expressing Staufen may be due to direct interference with dynein function (Januschke et al., 2002). Moreover, Mohr et al. (2021) found that the SRSs that are not proximal to the TAS in the *oskar* sequence also contribute to posterior localization of the mRNA in a Staufen-dependent manner. Although it is possible that these elements are close enough to the binding site for Egl-BicD-dynein-dynactin in the folded RNA molecule to interfere directly with the assembly or activity of the complex, they could also regulate *oskar* mRNA distribution through an independent mechanism. Other observations in our study do hint at other roles of Staufen. We observed that when Staufen is depleted, motile *oskar* RNPs tend to have a lower RNA content, suggesting an additional function of the protein in RNA oligomerization. Furthermore, whilst the magnitude of the effect was much smaller than for minus-end-directed motion, there was increased plus-end-directed velocity of a subset of *oskar* RNPs in ex vivo motility assays when Staufen was disrupted. Although this observation could reflect dynein's ability to promote kinesin-1 activity (Hancock, 2014), it is also possible that Staufen directly tunes the activity of the plus-end-directed motor.

An important question raised by our analysis of Staufen's effects on *oskar* mRNA transport was how the timing and location of this process is controlled. We provide evidence that this is based on another mRNA localization process in which Egl, as part of *stau* RNPs, is responsible for the enrichment of *stau* mRNA in the developing oocyte. We propose this mechanism constitutes a feed-forward type of switch, whereby the activity of the dynein-mediated transport machinery deploys its own negative regulator to a distant location. The resultant increase in Staufen levels in the ooplasm prevents dynein-mediated transport of *oskar* to the anterior of the oocyte, while the almost complete absence of the protein from the nurse cells is likely to be important for uninterrupted transport of *oskar* RNPs into the oocyte by dynein during early- and mid-oogenesis. Presumably, *stau* translation is suppressed during transit into the oocyte or the protein is translated en route but only builds up to meaningful levels where the RNA is concentrated in the oocyte. Staufen protein might also modulate its own localization in the ooplasm by antagonizing the association of *stau* mRNA with Egl,

and thereby its minus-end-directed motility. Consistent with this notion, the level of Egl on *stau* RNPs declines at a similar stage to when Staufen inhibits the association of Egl with *oskar* RNPs. As Staufen controls localization and translation of other mRNAs in the maturing oocyte (St Johnston et al., 1991; Ferrandon et al., 1994), mRNA-based regulation of the protein distribution in the egg chamber is likely to have functions that extend beyond orchestrating *oskar* mRNA localization. Given the functional conservation of Staufen protein (Heber et al., 2019) and the observation that the mRNA encoding mStau2 is localized in dendrites of mammalian neurons (Zappulo et al., 2017), it is plausible that the feed-forward loop established by *stau* RNA localization during *Drosophila* oogenesis is an evolutionarily conserved process that controls RNA trafficking and protein expression in polarized cells.

## Materials and methods

### Fly strains

To knock down *stau* and *egl* RNA levels, we used the P{TRiP.GL01531} (FBal0281234) and the P{TRiP.HM05180} (FBal0240309) transgenic lines. A TRiP line against the *w* gene (P{TRiP.GL00094}–FBal0262579) was used as a negative control. The following mutant lines were used to disrupt or modify gene function: *stau*[D3] (FBal0016165) and *stau*[R9] (FBal0032815); *egl*[1] (FBal0003574) and *egl*[2] (FBal0003575); *BicD*[1] (FBal0001140) and *BicD*[2] (FBal0001141); and *osk*[A87] (FBal0141009) and *Df(3R)p-XT103* (FBab0002842; a chromosomal deficiency that removes the *oskar* locus). *w*[1118] (FBal0018186) was used as the wild-type control.

To overexpress Staufen, we used the αTub67C:GFPm6-Staufen (FBal0091177), αTub67C:RFP-Staufen (Zimyanin et al., 2008) transgenic lines and a P{UASp-Staufen} transgene inserted onto the third chromosome (kind gift of F. Besse, Institut de Biologie Valrose, Nice, France).

To overexpress unlabeled Egl protein, we used the P{UASp-egl.B} transgene (FBtp0022425) inserted on the X or the 3rd chromosome. To label proteins of interest, we used the following fluorescently tagged reporter lines: P{Staufen-GFP} inserted on the 3rd chromosome, expressing Staufen-GFP under control of the endogenous *stau* promoter (kind gift of F. Besse); P{tub-egl.EGFP} (FBtp0041299) inserted on the 3rd chromosome, driving expression of Egl-GFP; P{UASp-BicD.eGFP} (FBtp0069955) and P{UAS-DCTN2-p50::GFP} (FBtp0016940) transgenic insertions to label BicD and p50/dynamitin, respectively. To drive the expression of the TRiP RNAi lines and other UASp constructs in the female germline, one copy of P{osk-Gal4} (FBtp0083699) inserted onto either the second or the third chromosome was used. For moderate overexpression of UASp-GFP-Staufen, we used one copy of P{matα4-GAL-VP16} (FBtp0009293) inserted onto the second chromosome.

To label *oskar* mRNPs in vivo and ex vivo, we used the *osk::oskMS2(10x)* and the *hsp83::MCP-EGFP* transgenes (Zimyanin et al., 2008) recombined on the third chromosome, and UASp-EB1-mCherry (a gift from D. Brunner, University of Zurich, Zurich, Switzerland) and *osk*-Gal4 transgenes recombined on the second chromosome. Their combination allows simultaneous

visualization of *oskar* mRNA (through binding of MCP-EGFP to *MS2* RNA repeats) and growing microtubule plus ends (through their binding to EB1-mCherry). The same *osk*-Gal4 transgene was used to drive the expression of the RNAi constructs (*stau* or *w* RNAi) in this background.

The endogenous Khc locus was tagged with the mKate2 coding sequence that was inserted in-frame, downstream of the Khc coding sequence (FBal0325536) (Gáspár et al., 2017).

The endogenous Dhc locus (*Dhc64C*) was tagged with the Emerald-GFP coding sequence to generate a GFP-Dhc expressing fly line according to protocols of the flyCRISPR website (Gratz et al., 2013; O'Connor-Giles et al., 2014; https://flycrispr.org/). The locus was targeted using CRISPR/Cas9 and a guide RNA targeting the following sequence: 5′-GAGTCACCCATGTCCCACAA-3′. The introduced double-stranded break was repaired by homologous recombination using an in-frame Emerald-GFP coding sequence flanked by two ∼700 bp long homology arms targeting around the Dhc translational initiation codon. F1 generation embryos were screened for GFP fluorescence to identify individuals with a modified genome. In-frame insertion was confirmed by sequencing the Dhc64C locus around the breakpoint. Flies homozygous for GFP-Dhc are viable and fertile. All stocks were raised on normal cornmeal agar at 25°C, and females were fed with wet yeast overnight before harvesting their ovaries.

### Ex vivo motility assay of *oskar* RNPs

The ex vivo motility analysis of *oskar* RNPs were carried out as described in Gaspar and Ephrussi (2017). Briefly, control and *stau* RNAi ovaries expressing EB1-mCherry and *oskMS2*-GFP were dissected and transferred onto silanized coverslips in a drop of BRB80 (80 mM PIPES, 1 mM $MgCl_2$, 1 mM EGTA pH 6.8). Several stage 9 egg chambers were isolated and pulled under Voltalef 10S halocarbon oil on the same coverslip. Under the oil, the nurse cells were removed by laceration using two fine tungsten needles. The isolated oocyte still in a "sack" of follicle cells were spatially separated from the remnants of nurse cell cytoplasm and, with a gentle prick on the oocyte anterior, the ooplasm was released onto the coverslip by a continuous slow pulling on the posterior of the oocyte-follicle "sack." Once several such preps were created, the ooplasmic extracts were imaged at 25°C using a Leica 7000 TIRF microscope with 100× oil (NA = 1.4) objective to visualize *oskMS2*-GFP and mCherry labeled microtubules excited with 488-nm and 561-nm diode lasers, respectively. Fluorescence was detected with a Photometrics Evolve 512 EM CCD camera with 140-nm lateral resolution behind a Chroma TRF89902 quad-band emission filter. Time-lapse image series were collected at 20 frames/s for ∼30 s and analyzed as described in Gáspár et al. (2017).

To analyze the relative mRNA content of *oskMS2*-GFP RNPs, a series of Gaussian functions was fitted to the GFP signal intensity distribution in each time-lapse series individually using the mixtools package in R (Benaglia et al., 2009; R Core Team, 2014). The smallest μ value of Gaussian fits was used to represent a single unit of RNA and each RNP was normalized to this value.

### RNA co-immunoprecipitation (RIP) from ovarian lysate

RIP was carried out as described in Gáspár et al. (2017). Briefly, ovaries from 50 flies were dissected in BRB80 and lysed in Pre-

XL buffer (20 mM Tris-Cl, 150 mM KCl, 4 mM MgCl$_2$, 2× cOmplete protease inhibitor cocktail [PIC], Roche, 1 mM PMSF pH 7.5; supplemented with 40 U of RiboLock RNase Inhibitor per 100 µl lysate). Ovaries were ground using a pestle and centrifuged for 1 min at 500 × g. The supernatant was extracted and crosslinked at 0.3 J/cm². The lysate was homogenized with 1 volume of Pre-XL buffer, 1 volume of RIPA buffer (10 mM Tris-Cl pH 7.5; 150 mM NaCl; 0,5 mM EDTA; 0,1% SDS; 1% Triton X-100; 1% Deoxycholate, 0.09% Na-Azide) buffer and eight volumes of low-salt buffer (20 mM Tris-Cl, 150 mM KCl, 0.5 mM EDTA, 1 mM PMSF). GFP-Trap_MA beads were washed with low-salt buffer and blocked for 60 min at room temperature in Casein Blocking Buffer (Sigma-Aldrich) supplemented with 50 µg/ml heparin (Sigma-Aldrich). Lysate was incubated with beads for 90 min at 4°C. The beads were then washed six times with high-salt buffer (20 mM Tris-Cl, 500 mM NaCl, 1 mM EDTA, 0.1% SDS, 0.5 mM DTT, 1xPIC, 1 mM PMSF) and twice with PBT (PBS + 0.1% Triton). Endogenous RNA cross-linked to bait protein was recovered from the beads using the QuickRNA Microprep Kit (Zymo Research). Complementary DNA was synthesized using the SuperScript III First-Strand Synthesis kit (Invitrogen) and used as template for PCR using *oskar* primers.

### Single-molecule fluorescent in situ hybridization (smFISH)

smFISH was carried out as described previously (Gáspár et al., 2017; Heber et al., 2019). Briefly, ssDNA oligonucleotides complementary to *oskar*, *bicoid* (Heber et al., 2019), and *stau* mRNAs (Table S1) were mixed and labeled with Atto532, Atto565, or Atto633 according to the protocol described in Gaspar et al. (2017).

*Drosophila* ovaries expressing fluorescent reporter transgenes were dissected in 2% paraformaldehyde (PFA; Thermo Fisher Scientific), 0.05% Triton X-100 in PBS and fixed for 20 min. The fixative was removed by two 5-min washes in PBT (PBS + 0.1% Triton X-100 pH 7.4). Ovaries were pre-hybridized in 200 µl 2 × HYBEC (300 mM NaCl, 30 mM sodium citrate pH 7.0, 15% ethylene carbonate, 1 mM EDTA, 50 µg/ml heparin, 100 µg/ml salmon sperm DNA [Invitrogen], 1% Triton X-100) for 10 min at 42°C. Meanwhile, the 50 µl probe mixture (5 nM per individual oligonucleotide) was prepared and pre-warmed to hybridization temperature. After 10 min of pre-hybridization, the probe mix was mixed into the pre-hybridization mixture. After 2 h of hybridization at 42°C, the unbound probe molecules were washed out of the specimen by two washes in pre-warmed HYBEC and a final wash in PBT at room temperature. To boost signal intensity for the protein enrichment analysis of Staufen-GFP in early egg chambers, we incubated the ovaries in GFP-Booster (Chromotek, labeled with AlexaFluor 488) diluted 1:2,000 in PBT for 1 h and washed the sample three times for 10 min in PBT. Ovaries were mounted in Vectashield for imaging.

### smFISH followed by immunofluorescence

5–10 pairs of ovaries were dissected in PBS and permeabilized for 20 min in 2% PFA in PBT. Fixed ovaries were washed three times in PBST for 10 min at room temperature. smFISH steps were performed as described above, with slight modifications. The ovaries were incubated with a final probe concentration of 1 nM/probe in 250 µl HYBEC solution at 37°C for 3 h. The ovaries were then washed twice in HYBEC solution for 15 min at 37°C to remove excess probes, and once with PBT at room temperature. The ovaries were blocked in 1× casein/PBT (diluted from 10X casein blocking buffer, Sigma-Aldrich) for 1 h and incubated with rabbit anti-Staufen antibody (St Johnston et al., 1991) diluted 1/1,000 in blocking buffer overnight at 4°C. The samples were then incubated with secondary antibody (goat anti-rabbit conjugated with Alexa-647; Jackson ImmunoResearch), diluted 1/1,000 in blocking buffer, for 1 h at room temperature. Three washes were performed in PBT (10 min each) at room temperature. Ovaries were mounted in 80% 2,2 thio-diethanol (TDE)/ PBS.

### Microscopy

Imaging of oocytes was carried out on a Leica TCS SP8 confocal laser scanning microscope using a 20× dry (NA = 0.75) objective for imaging the RNA distribution in the oocytes and a 63× oil immersion (NA = 1.4) objective for analysis of RNA or Staufen-GFP protein enrichment in early oocytes and for colocalization analysis between RNPs and the fluorescently labeled Staufen and Egl proteins. Dyes and fluorescent proteins were excited sequentially at their respective excitation maxima with a tunable white light laser and fluorescence was detected by HyD detectors behind acousto-optic tunable filters (AOTFs) adjusted to the respective dye emission maxima: GFP/AlexaFluor 488: 488 nm, 500–545 nm; RFP: 594 nm, 610–650 nm, Atto 532: 535 nm, 545–575 nm; Atto 565: 561 nm, 575–620 nm, Atto 633: 634 nm, 650–750 nm. AlexaFluor 647: 645 nm, 670–750 nm.

For Fig. 2, A–G and Fig. S2, imaging of reconstituted RNPs by TIRF microscopy was performed at room temperature with 100 ms exposures at ~2 frame/s using a customized Nikon Nikon TIRF system (Nikon Ti2 base, Nikon 100× Apo TIRF NA = 1.49 oil immersion objective, Andor iXon$^{EM}$+ DU-897E EMCCD camera) controlled by Micro-manager acquisition software (Edelstein et al., 2014). AlexaFluor 488-*oskar* RNA was imaged using a Coherent Sapphire 488 nm laser line, TMR-dynein was imaged using a Coherent Sapphire 561 nm laser line, and Cy5-*oskar* RNA and AlexaFluor 647-dynein were imaged using a Coherent CUBE 647 nm laser line. For data presented in Fig. 2, H and I and Fig. S2 D, movies were acquired with 100-ms exposures at 4 frames/s for a total of 3 min at room temperature using Nikon Elements acquisition software on a Nikon Eclipse Ti2 inverted TIRF system equipped with a Nikon 100× Apo TIRF oil immersion objective (NA = 1.49) and Photometrics Prime 95B CMOS camera. TMR-dynein was imaged using the 561-nm laser line with a Chroma ET-561 nm Laser Bandpass Set (ET576lp and ET600/50m) and ET605/52m emission filter.

### RNA distribution analysis

Analysis of *oskar* mRNA distribution was carried out as described in Gaspar et al. (2014); Gáspár et al. (2017); and Heber et al. (2019). Briefly, we manually defined the outlines of the oocytes and their anteroposterior (AP) axis using a custom-made ImageJ plug-in; the smFISH signal was then redistributed into a 100 × 100 matrix. Each column of the matrix represents the relative amount of signal found under 1% of the AP axis length,

with anterior on the left (column 1) and posterior on the right (column 100). The matrices from different oocytes of the same genotype and stage were averaged to obtain an average view RNA localization pattern. The center-of-mass (relative to the geometric center of the oocyte) was determined and compared statistically using a Kruskal–Wallis test followed by pairwise Mann–Whitney U test against the control.

### RNA and protein enrichment analysis

The boundaries of the somatic follicle cells, nurse cells, and oocyte were manually defined, and the fluorescence intensity of smFISH or GFP signal in these compartments was measured by ImageJ. The extent of enrichment of RNA and protein in the somatic follicle cells and the oocyte was obtained by normalizing the measured fluorescence intensity values to the corresponding values obtained for the nurse cells.

### Colocalization analysis between RNPs and fluorescent reporters

Analysis was carried out as described in Gáspár et al. (2017) and Heber et al. (2019). Briefly, images for colocalization analysis were deconvolved using Huygens Essential (https://svi.nl) and segmented using a custom ImageJ (Schneider et al., 2012) plug-in. Nearest-neighbor pairs between RNPs and fluorescent reporters were identified, and their position and signal intensities were extracted from the nurse cell and oocyte compartments, excluding any nuclear areas in the field of view. Quantification of mRNA copy number per RNP and normalization of fluorescent reporter signal was carried out by fitting multiple Gaussian functions to the corresponding signal intensity distributions taken from the nurse cells using the mixtools package in R (Benaglia et al., 2009; R Core Team, 2014). The μ value of Gaussian fit that described the largest portion of the distribution in the nurse cells (almost the lowest value of all fitted μ values) was taken as the signal intensity of one unit (for RNPs the intensity of a signal mRNA molecule). These unit values were used to normalize raw signal intensities. RNPs were clustered based on this normalized intensity under the following rule: $(2^i:2^{i+1})$, i ∈ [0:8], i.e., 1, 2:3, 4:7, 8:15, etc. The observed nearest-neighbor colocalization frequencies were computed for each cluster and were compared to the expected colocalization frequencies (governed by the object-densities determined in randomized object localizations). The reported colocalization frequencies were corrected by subtracting the expected values from the observed values, which could result in negative frequencies, i.e., depletion of a molecule from *oskar* mRNPs. To estimate the 95% confidence intervals in each cluster, RNPs were randomly grouped together in groups of 50, then mean and 95% confidence interval of corrected colocalization values calculated. True difference from zero was established by one sample *t* tests (normality of the distribution was assumed due to random sampling). Similarly, the mean, normalized intensity of colocalizing fluorescent reporter molecules was calculated for each cluster. Correlation between RNA content and the normalized mean fluorescent reporter intensity was tested and compared using least-squares means analysis in R (Lenth, 2016). We typically analyzed 5,000–40,000 *oskar* RNPs (100–800 groups) and

2,000–4,000 *stau* RNPs (40–80 groups) imaged in 4–6 egg chambers collected from 3–4 females per condition.

### Statistical analysis

Statistical analyses were performed as indicated in the figure legends using R (R Core Team, 2014), RStudio (www.rstudio.com), and GraphPad Prism version 9.1.1 for MacOS (GraphPad Software, www.graphpad.com). All graphs were plotted by the ggplot2 library in R (Wickham, 2016) and GraphPad Prism version 9.1.1 for MacOS (GraphPad Software, www.graphpad.com). For analyses in which parametric statistical tests were used, the normality of the data distribution was assumed but not formally tested.

### Protein expression, purification, and fluorescent labeling

Recombinant dynein (human), Egl (*Drosophila*), BicD (*Drosophila*), and BicD2N (mouse) were expressed, purified, and fluorescently labeled as described previously (Hoang et al., 2017; McClintock et al., 2018) using the MultiBac *Sf*9 insect cell expression system (Vijayachandran et al., 2013). Complexes of the complete dynein holoenzyme (Dynein heavy chain—DYNC1H1 in pAceBac1; Dynein intermediate chain—DYNC1/2, Dynein light intermediate chain—DYNC1LI2, Tctex—DYNLT1, Robl—DYNLRB1, LC8—DYNLL1 in pIDC), the coexpressed Egl/BicD complex (Egl isoform B: NM_166623 in pAceBac1; BicD: NM_165220 in pIDC), and BicD2N (BICD2: NM_001039179 in pAceBac1) were expressed in *Sf*9 insect cells and purified by affinity chromatography using ZZ-tags on the N-terminus of Dynein heavy chain, C-terminus of Egl, or N-terminus of BicD2N, respectively. *Sf*9 cells were lysed by dounce homogenization in lysis buffer (50 mM HEPES pH 7.3, 100 mM [dynein, BicD2N] or 500 mM [Egl/BicD] NaCl, 10% glycerol, 1 mM DTT, 0.1 mM MgATP, 1× EDTA-free cOmplete protease inhibitor cocktail [Roche], 2 mM PMSF) and the lysate was clarified by ultracentrifugation (Beckman Type 70 Ti rotor) at 361,000 × *g* (at $r_{average}$) before application to IgG-Sepharose beads for 3 h at 4°C. The beads were collected by gravity flow column and subsequently washed twice with five column volumes of lysis buffer and twice with five column volumes of TEV buffer (50 mM Tris-HCl pH 7.4, 150 mM KOAc, 2 mM MgOAc, 1 mM EGTA-KOH pH 7.3, 10% glycerol, 0.1 mM MgATP, 1 mM DTT). TEV protease was applied to the washed beads overnight with gentle rolling at 4°C and the protein eluted by gravity flow. Fluorescent labeling of the Dynein heavy chain and BicD2N with SNAP-Cell TMR Star or SNAP-Surface Alexa Fluor 647 via N-terminal SNAPf tags was performed either on-column prior to washing 3× with five column volumes TEV buffer and elution from the IgG affinity resin (dynein, Egl/BicD) or in solution following TEV-cleavage (BicD2N). Excess dye was removed from BicD2N by passing through a PD10 desalting column. All complexes were further purified by FPLC-based gel-filtration chromatography, using either TSKgel G4000SWxl (dynein, BicD2N) or Superose 6 Increase 3.2/300 GL (Egl/BicD) columns equilibrated in GF150 buffer (25 mM HEPES pH 7.3, 150 mM KCl, 1 mM MgCl$_2$, 5 mM DTT, 0.1 mM MgATP, 10% glycerol). Eluted protein was concentrated to ~1 μM (dynein), ~10–15 μM (BicD2N), or ~2–3 μM (Egl/BicD) using an Amicon Ultra-4 100 kD MWCO concentrator.

The dynactin complex was purified natively from pig brains as described previously (Schlager et al., 2014; Urnavicius et al., 2015). Brain extracts were made by washing the brain tissue and removing meninges and blood vessels prior to homogenization in lysis buffer (35 mM PIPES-KOH pH 7.2, 5 mM $MgSO_4$, 1 mM EGTA-KOH pH 8.0, 0.5 mM EDTA pH 8.0, 1 mM DTT, 1× EDTA-free cOmplete protease inhibitor cocktail [Roche], 2 mM PMSF). Extracts were clarified by centrifugation and fractionated by FPLC using SP Sepharose (cation exchange, ~250 ml bed volume), in 35 mM PIPES-KOH pH 7.2, 5 mM $MgSO_4$, 1 mM EGTA-KOH pH 8.0, 0.5 mM EDTA pH 8.0, 1 mM DTT, 0.1 mM MgATP across a linear 0–1 M KCl gradient. Fractions containing dynactin were pooled and further fractionated on a Mono Q 4.6/100 PE anion exchange column in 35 mM PIPES-KOH pH 7.2, 5 mM MgSO4, 1 mM EGTA-KOH pH 8.0, 0.5 mM EDTA pH 8.0, 1 mM DTT across a linear 0–1 M KCl gradient. Dynactin-containing fractions were pooled and concentrated in an Amicon Ultra-4 100 kD MWCO concentrator before application to a TSKgel G4000SW gel-filtration) column equilibrated in GF150 buffers. Eluted protein was concentrated as before to ~2 µM.

Recombinant Staufen (*Drosophila*) was cloned as fusion protein with an N-terminal $His_6$-rsEGFP2-tag and a C-terminal $His_6$-tag in pFastBacDual between the KpnI and HindIII sites and expressed in *Sf*21 insect cells using the Bac-to-Bac expression system (Thermo Fisher Scientific). The protein was purified by affinity chromatography on HisTrap FF in 1× PBS + 880 mM NaCl, 400 mM arginine, 10–200 mM imidazole, 2 mM DTT and HiTrap Heparin HP in 40 mM Bis-Tris pH 7.5, 150 mM NaCl—1 M NaCl, 40 mM arginine, 2 mM DTT with subsequent filtration on Superdex200 Increase in a final buffer of 40 mM Bis-Tris pH 7.5, 150 mM NaCl, 2 mM DTT.

In all cases, eluted protein was dispensed into single-use aliquots, flash-frozen in liquid nitrogen, and stored at –80°C.

### RNA synthesis and purification

The *oskar* RNA used for the reconstitution assay was uncapped and synthesized in vitro using the MEGAscript T7 Transcription Kit (Ambion). The RNA was transcribed from a linearized plasmid DNA template containing an ~529-nt region of the *osk* 3′UTR previously defined as region 2 + 3 that includes the oocyte entry signal (OES) and promotes the localization of reporter transcripts to the developing oocyte (Jambor et al., 2014). For fluorescent labeling of transcripts, Cy5 UTP or ChromaTide Alexa Fluor 488 UTP was included in the transcription reaction with a two or fourfold excess of unlabeled UTP, yielding transcripts with an average of ~5 or ~14 fluorophores per RNA molecule, respectively. Excess fluorescent UTP was removed using two successive rounds of Sephadex G-50 desalting spin columns per transcription reaction. Transcripts were subsequently purified by ammonium acetate/ethanol precipitation and resuspension in nuclease-free water. RNA was dispensed into single-use aliquots and stored at –80°C.

### In vitro RNP motility assay with purified proteins

TIRF-based single-molecule reconstitution assays were performed as described previously (McClintock et al., 2018). Taxol-stabilized porcine microtubules were polymerized with a mixture of unlabeled tubulin, HiLyte 488 tubulin, and biotin-conjugated tubulin, and immobilized in imaging flow chambers by streptavidin-based linkages to biotin-PEG passivated cover slips. Assembly mixes containing relevant combinations of dynein, dynactin, Egl, BicD, BicD2N, *oskar* RNA, and Staufen were then diluted to concentrations suitable for imaging of single molecules (~2.5 nM for dynein), applied to the flow chamber and subsequently imaged by TIRF microscopy. For assays testing the effect of Staufen on RNA or dynein motility, the complexes were assembled for 45–60 min on ice with components diluted in GF150 buffer to concentrations of 100 nM TMR-dynein, 200 nM dynactin, 500 nM Egl/BicD or BicD2N and 1 µM *oskar* RNA. This assembly mix was then diluted 1:1 with either ~4 µM rsEGFP2-Staufen or Staufen storage buffer (40 mM Bis-Tris pH 7.5, 150 mM NaCl, 2 mM DTT) and incubated for at least a further 15 min on ice. Just prior to application to the imaging chamber, this mixture was further diluted 20-fold in Motility Buffer (30 mM HEPES pH 7.3, 50 mM KCl, 5 mM $MgSO_4$, 1 mM EGTA pH 7.3, 1 mM DTT, 0.5 mg/mLl BSA, 1 mg/ml α-casein, 20 µM taxol) with added MgATP (2.5 mM final concentration) and 1× oxygen scavenging system (1.25 µM glucose oxidase, 140 nM catalase, 71 mM 2-mercaptoethanol, 25 mM glucose final concentrations) to reduce photobleaching. For assays testing the effect of Egl/BicD on RNA binding and motility on microtubules, the complexes were assayed as above except that primary assembly mixes consisted of 100 nM Alexa Fluor 647-dynein, 200 nM dynactin, 500 nM Egl/BicD or Egl/BicD storage buffer (GF150), and 1 µM *oskar* RNA, which were subsequently diluted 1:1 in Staufen storage buffer and processed as above. Binding and motility of RNA and dynein in the in vitro reconstitution assays were manually analyzed by kymograph in FIJI (Schindelin et al., 2012) as described previously (McClintock et al., 2018) with microtubule binding events defined as those lasting for a minimum of three continuous frames of acquisition and processive events defined as those that exhibited unidirectional movement of at least 5 pixels (525–550 nm depending on the microscope system) during this time, regardless of velocity. For Fig. S2 B, values for the frequency of RNA binding to microtubules were corrected for non-specific RNA binding to the coverslip by generating kymographs in regions that were completely devoid of microtubules and equivalent in size to the median length of all microtubules analyzed in each field of view. The average microtubule binding frequency for RNA within these background regions was subtracted from the average RNA binding frequency of each analyzed microtubule to derive the corrected value.

### Online supplemental material

Fig. S1 shows that *oskar* mRNA localization and transport are impaired in *stau* mutants. Fig. S2 shows that both Egl and BicD are required for formation of transport-competent dynein-RNA complexes in vitro. Fig. S3 quantitatively shows how (over)expression of Staufen suppresses *oskar* mislocalization defects. Fig. S4 supports Fig. 4 in showing that Staufen and Egl associate with *oskar* RNPs and that Staufen interferes with Egl association. Fig. S5 shows the distribution of *oskar* mRNPs along the full length of the anteroposterior axis of stage 9 oocytes. Fig. S6 shows the

localization of Staufen protein and *stau* mRNA in the developing egg chambers and the effects of overexpression of Staufen on mRNA transport and oogenesis. Table S1 shows the sequences of the oligonucleotides used to perform smFISH analysis of *stau* mRNA.

## Data availability

All raw imaging data are available for re-analysis purposes upon request from I. Gaspar (imre.gaspar@turbine.ai) or S.L. Bullock (sbullock@mrc-lmb.cam.ac.uk).

## Acknowledgments

We thank the EMBL Advanced Light Microscopy and Gene Core Facilities for their support. We acknowledge Alessandra Reversi (EMBL *Drosophila* Injection Service) for fly transgenesis and thank Florence Besse (Institute de Biologie Valrose, Nice, France) and Damian Brunner (University of Zurich, Zurich, Switzerland) for *Drosophila* strains and Dierk Niessing (University of Ulm, Ulm, Germany) for reagents for recombinant Staufen expression.

Work in S.L. Bullock's group is supported by the Medical Research Council, as part of United Kingdom Research and Innovation (also known as UK Research and Innovation; MRC file reference number MC_U105178790). M.A. McClintock is supported by a BBSRC project grant (BB/T00696X/1). L.J. Phea was supported by Deutsche Forschungsgemeinschaft (DFG) Forschergruppe grants EP37/2-1 and EP37/4-1 to A. Ephrussi. A. Ephrussi gratefully acknowledges the support of the European Molecular Biology Laboratory. For the purpose of the MRC open access policy, the authors have applied a CC BY public copyright license to any Author Accepted Manuscript version arising. Open Access funding was provided by the European Molecular Biology Laboratory.

Author contributions: I. Gaspar and M.A. McClintock designed experiments. I. Gaspar, L.J. Phea., M.A. McClintock. and S. Heber carried out the experiments and analyzed the data. S.L. Bullock and A. Ephussi supervised the work. All authors discussed the data and contributed to manuscript preparation.

Disclosures: All authors have completed and submitted the ICMJE Form for Disclosure of Potential Conflicts of Interest. A. Ephrussi reported grants from Deutsche Forschungsgemeinschaft (DFG) during the conduct of the study. No other disclosures were reported.

Submitted: 27 January 2023

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

# Supplemental material

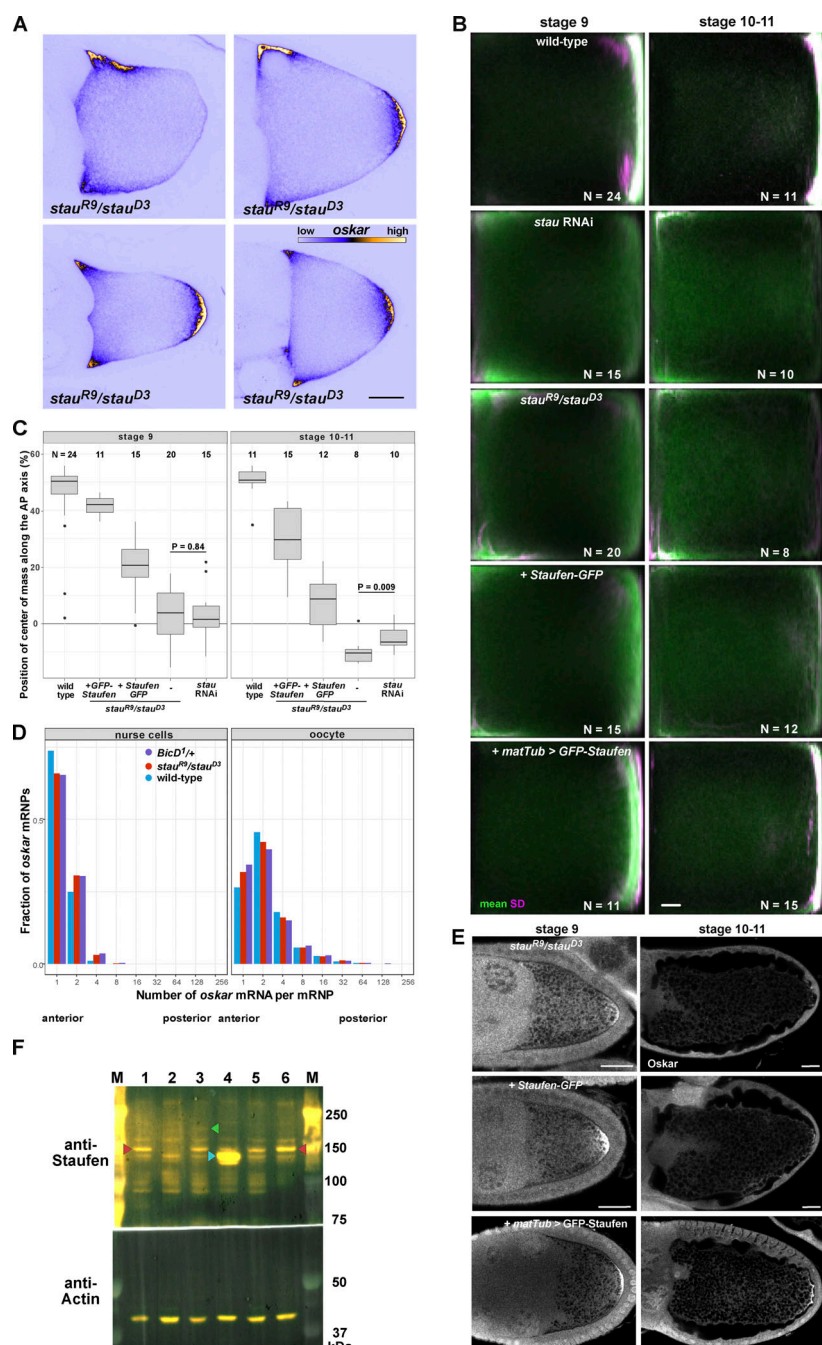

**Figure S1.** **Expression of the different Staufen transgenes and their effects on *oskar* mRNA localization. (A)** *oskar* mRNA localization (shown in blue/yellow) in stage 9 *stau^R9^/stau^D3^* mutant oocytes. Scale bars represent 20 μm. **(B)** Mean (green) and variance (SD, magenta) of *oskar* mRNA distribution in stage 9 and stage 10–11 oocytes. Numbers indicate the number of oocytes analyzed for each condition and scale bar represents 10% of anteroposterior axis length. Anterior is to the left, posterior is to the right. The three panels in the top left (stage 9 wild-type, *stau* RNAi and *stau^R9^/stau^D3^*) are reused in Fig. S3. **(C)** Position of *oskar* center-of-mass along the AP axis in stage 9 and stage 10–11 oocytes for each condition. 0 is the geometric center of the oocyte, with the posterior pole located at 58%. **(D)** *oskar* mRNA content distribution in wild-type (blue), *stau* null (red) and dominant *BicD^1^* mutant (purple) nurse cells and oocytes. In the nurse cells, most *oskar* RNPs contain 1–2 copies of the RNA, consistent with previous reports (Little et al., 2015). **(E)** Oskar protein expression in Staufen null oocytes coexpressing transgenic Staufen-GFP or GFP-Staufen. Scale bars represent 20 μm. **(F)** Western blot detection of Staufen in wild-type (lane 1), *stau^R9^/stau^D3^* (2), Staufen-GFP (3), matTub > GFP-Staufen (4), *stau* RNAi (5), and control RNAi (6) ovarian lysates. Endogenous Staufen (~150 kD) indicated with red arrowhead. Staufen-GFP (lane 3) migrates slower than the untagged protein (green arrowhead), while matTub > GFP-Staufen migrates faster due to an N-terminal truncation of Staufen (blue arrowhead; Micklem et al., 2000). Note the similar distribution of *oskar* in *stau^R9^/stau^D3^* and in *stau* RNAi oocytes (B and C), despite residual Staufen expression in Staufen RNAi ovarian lysates (F, lanes 5 and 6). The overexpressing GFP-Staufen (F, lane 4) transgene largely rescues *oskar* mislocalization (B and C) and Oskar protein expression defects (E) observed in *stau* null mutants. The Staufen-GFP transgene, expressed at low levels (I, lane 3), rescues *oskar* mRNA localization at stage 9, but the RNA is not maintained at the posterior at stages 10–11 (B and C), likely due to insufficient Oskar protein expression at the posterior (E), which is essential for *oskar* mRNA anchoring at the oocyte posterior during the later stages (Vanzo and Ephrussi, 2002). Source data are available for this figure: SourceData FS1.

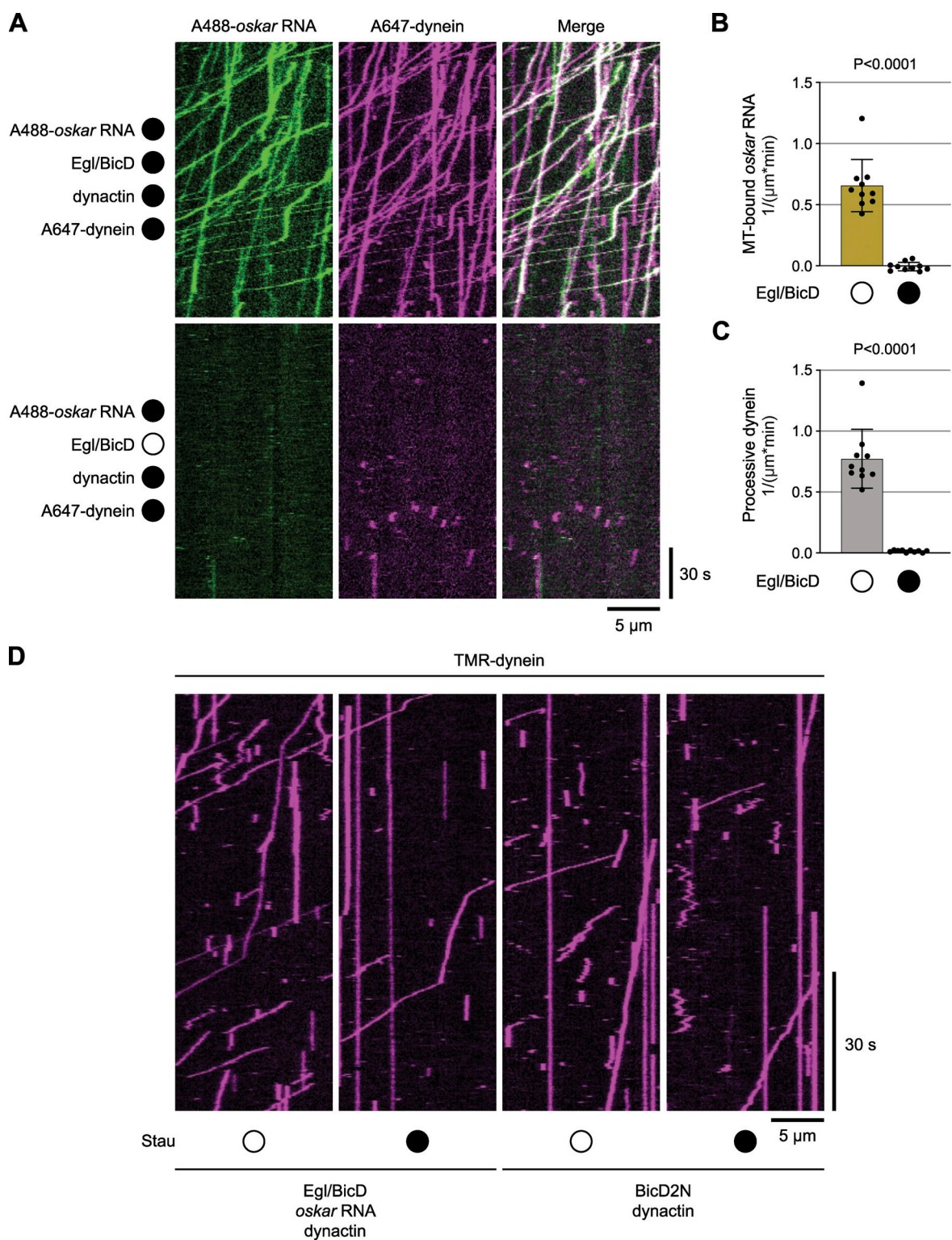

Figure S2. **Egl and BicD are required for formation of transport-competent dynein-RNA complexes in vitro. (A)** Example kymographs (time-distance plots) showing behavior of dynein and *oskar* RNA in the presence and absence of Egl and BicD; in each condition, dynactin is also present but not fluorescently labeled (filled and empty circles represent the presence and absence of indicated proteins, respectively). Egl and BicD were co-expressed and co-purified (see Materials and methods). **(B and C)** Charts showing the total number of microtubule (MT) binding events for *oskar* RNA (B) and number of processive dynein complexes on microtubules under conditions shown in A. In B, values were corrected for non-specific background binding of *oskar* RNA to the imaging surface as described in Materials and methods. Plots show the mean ± SD of values from 10 individual microtubules (represented by black circles) derived from 183 to 918 single RNA particles (B) or from 384 to 1,725 single dynein particles (C) per condition. Statistical significance and P-values were determined with Mann–Whitney tests. **(D)** Example kymographs (time-distance plots) showing the behavior of dynein activated by *oskar* RNA, Egl/BicD, and dynactin or by BicD2N and dynactin in the presence (filled circle) and absence (open circle) of Staufen. Quantification of these data is presented in Fig. 2, H and I.

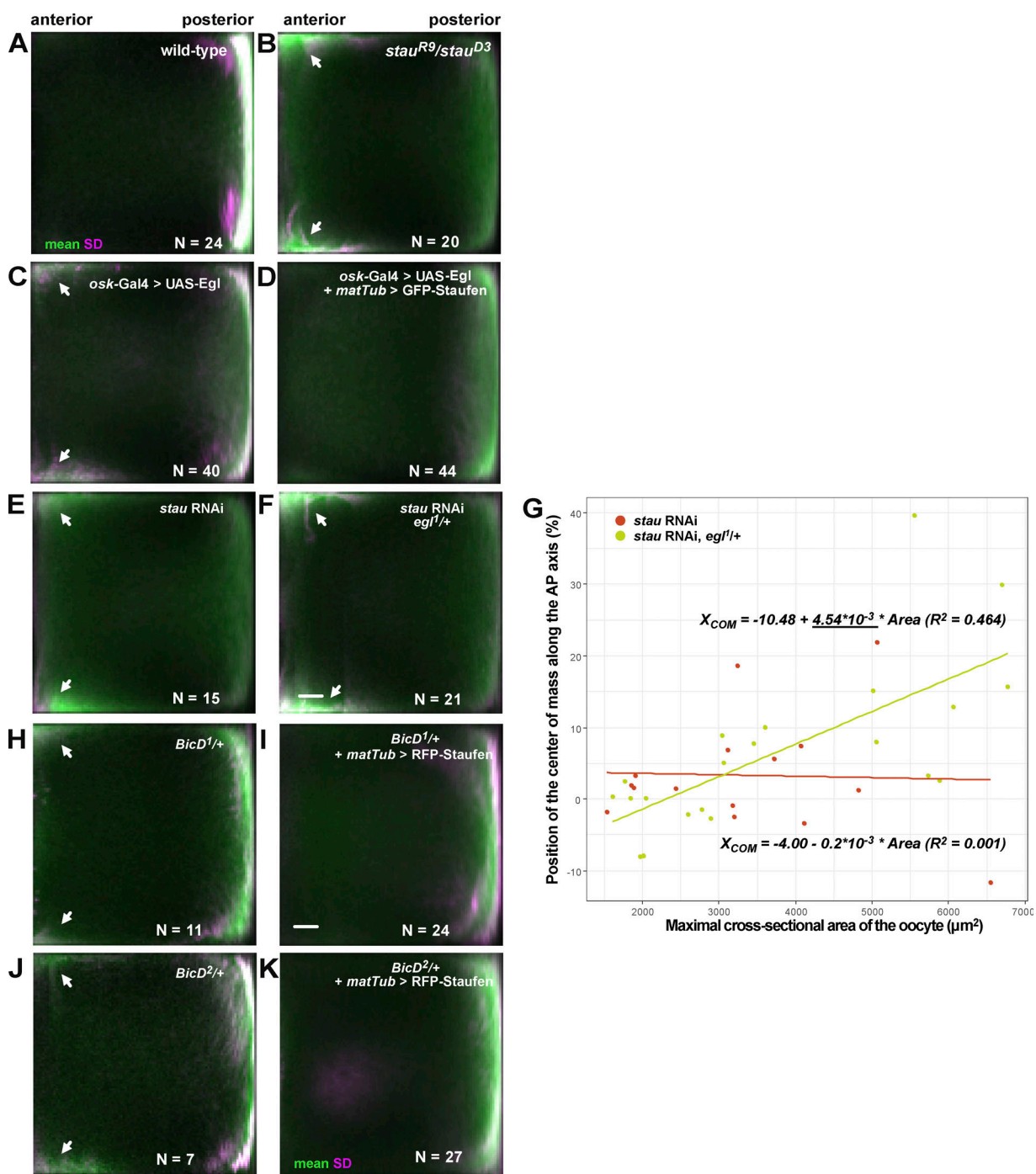

**Figure S3.** **Suppression of *oskar* mislocalization. (A–F)** Average distribution of *oskar* mRNA (green) and variability of the distribution (SD, magenta) in stage 9 oocytes of the indicated genotypes. N indicates the number of oocytes analyzed. Scale bar represents 10% of anteroposterior axis length. Anterior is to the left, posterior to the right. Arrowheads indicate the ectopic localization of *oskar* mRNA at the anterior cortex. The A, B, and E panels (stage 9 wild-type, *stauR9*/*stauD3* and *stau* RNAi) are reused in Fig. S1. **(G)** Distribution of the observed *oskar* center-of-mass in stage 9 *stau* RNAi oocytes in the presence of two (red) or one (yellow) functional copies of *egl* as a function of oocyte size, used here as a proxy for developmental stage. Solid lines show the best linear fits to the data. The equation and the square of the goodness-of-fit ($R^2$) are indicated. Such moderate rescue was expected as *oskar* RNPs entering the oocyte are thought to be associated with Egl. Note that there is no significant linear correlation between oocyte size (developmental stage) and the position of *oskar* mRNA center-of-mass in *stau* RNAi (red), indicating an *oskar* mislocalization phenotype. There is a moderate correlation with a significant slope (underlined) when one copy of *egl* is removed (yellow), indicating progressive posterior localization of *oskar* mRNA at stage 9. **(H–K)** Average distribution of *oskar* mRNA (green) and variability of the distribution (SD, magenta) in stage 9 oocytes of the indicated genotypes. N indicates the number of oocytes analyzed. Scale bar represents 10% of anteroposterior axis length. Anterior is to the left, posterior to the right. Arrowheads indicate the ectopic localization of *oskar* mRNA at the anterior cortex.

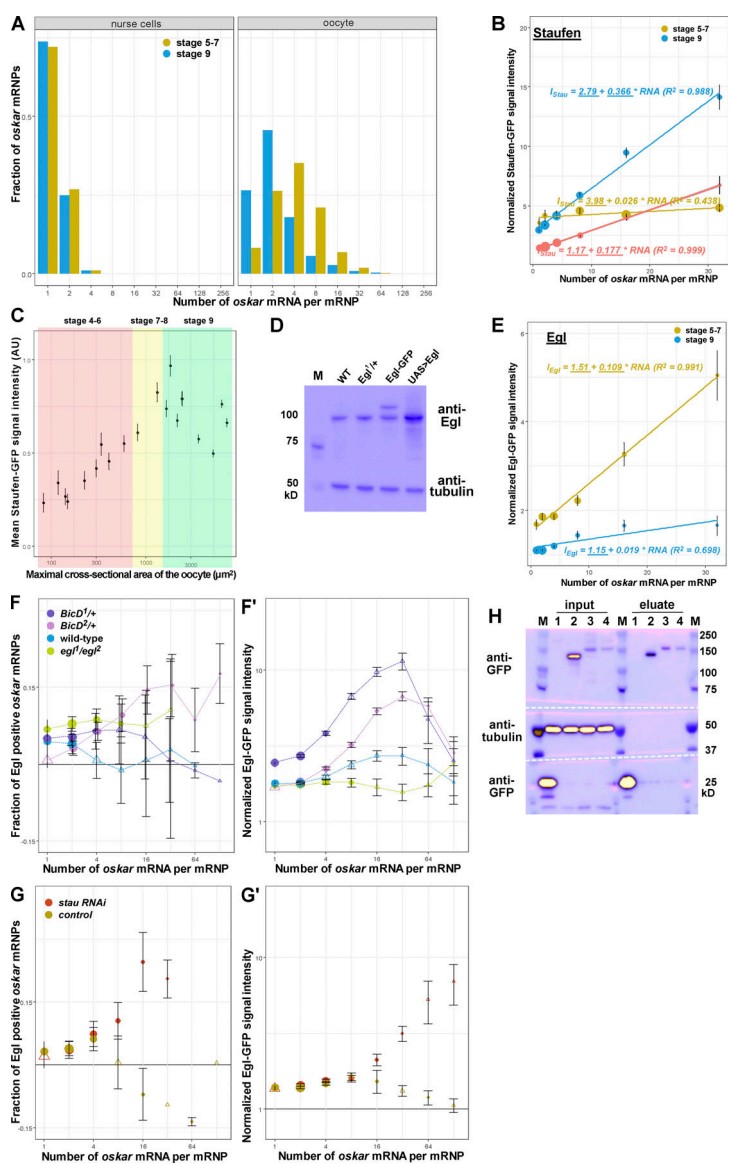

Figure S4. **Staufen and Egl association with *oskar* RNPs. (A)** *oskar* mRNA content distribution in wild-type nurse cells and oocytes at stages 5–7 (yellow) and 9 (blue) of oogenesis. In the nurse cells, most *oskar* RNPs contain 1–2 copies of the RNA, consistent with previous reports (Little et al., 2015). *oskar* RNP content increases in the oocyte: at stages 5–7 most RNPs contain 4+ copies of the RNA, which decreases to >2 copies during *oskar* posterior localization (stage 9; Little et al., 2015). **(B)** Normalized Staufen-GFP signal intensity as a function of *oskar* mRNA content at stages 5–7 (yellow) and stage 9 (blue and red). Staufen-GFP signal intensity was measured in the complete absence (yellow and blue) or presence (red) of endogenous, unlabeled Staufen. Fitted linear models showing the correlation between Staufen-GFP signal intensity and *oskar* mRNA copy number as solid lines and equations. Underscored parameters of the models are significantly different from zero (P < 0.05). The slopes of the two fitted models are significantly different (P < 0.0001, ANOVA). **(C)** Mean signal intensity of Staufen-GFP measured at multiple locations throughout developing oocytes. Size of the oocytes (x-axis) is used as a proxy of developmental time and, along with morphological features, for staging of the oocytes (shaded areas as indicated in the panel). **(D)** Western blot showing Egl protein detected by anti-Egl antibody in the indicated genotypes. Tubulin was used as a loading control. **(E)** Normalized Egl-GFP signal intensity as a function of *oskar* mRNA content at stages 5–7 (yellow) and stage 9 (blue). Fitted linear models showing the correlation between Egl-GFP signal intensity and *oskar* mRNA copy number as solid lines and equations (top—stages 5–7, bottom—stage 9). Underscored parameters of the models are significantly different from zero (P < 0.05). The slopes of the two fitted models are significantly different (P < 0.0001, ANOVA). **(F–G′)** Association of Egl-GFP with *oskar* RNPs in oocytes with *BicD1* (purple) or *BicD2* (pink) alleles (F and F′) or expressing *stau* (red) or control (brown) RNAi (G and G′). Note that knock-down of Staufen results in similar retention of Egl on *oskar* RNPs as in the complete absence of Staufen protein (Fig. 4, E and E′). In E, G, and G′, egg chambers expressed a single copy of Egl-GFP in the presence of two endogenous wild-type egl alleles, except in the case of the rescued egl mutants (G, *egl1/egl2*, green). Although we observed a slightly elevated fraction of Egl positive RNPs when unlabeled Egl was absent (*egl1/egl2*, green), larger RNPs containing 16+ copies of *oskar* mRNA displayed no significant association with Egl (G) and the relative amounts of Egl on *oskar* RNPs were identical to what was observed in the presence of endogenous, unlabeled Egl (G′, blue). 14,321–43,299 *oskar* RNPs per genotype were analyzed. Triangles indicate that the fraction of GFP-positive *oskar* RNPs is not significantly different from zero (P > 0.01, one sample t test). In F, datapoints are slightly offset in the x-axis to facilitate comparison. **(H)** Western blot of input lysates and eluates after RNA immunoprecipitation in the presence (lane 3) or the absence (lane 4) of Staufen. Bait proteins—monomeric EGFP (lane 1), GFP-Staufen (lane 2) and Egl-GFP (lanes 3,4)—are detected by anti-GFP antibody. Anti-tubulin staining was used to monitor potential contamination of the eluates. In D and H, blue and yellow indicate low and high intensity of signal, respectively. Source data are available for this figure: SourceData S4.

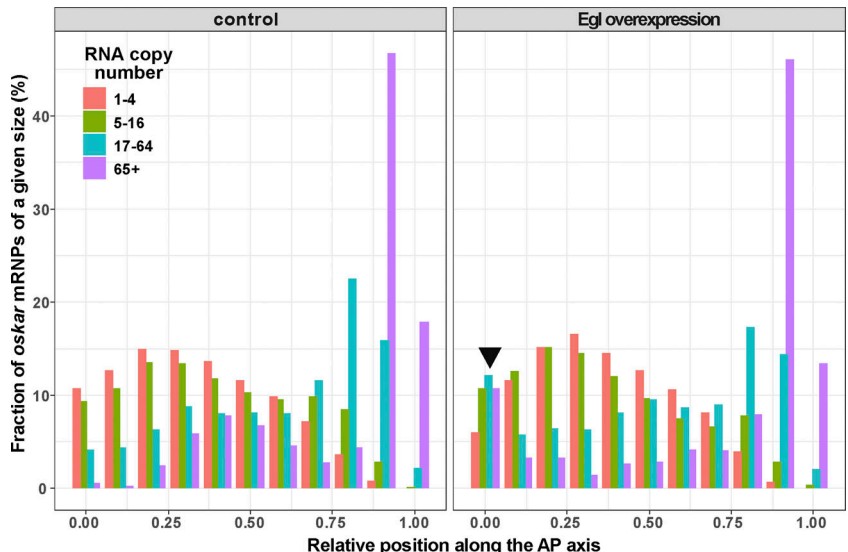

Figure S5. **Localization of *oskar* RNPs along the anteroposterior axis.** Relative distribution of *oskar* RNPs grouped by RNA content along the antero-posterior axis in wild-type and in Egl overexpressing (*osk-Gal4>UAS-Egl*) oocytes.

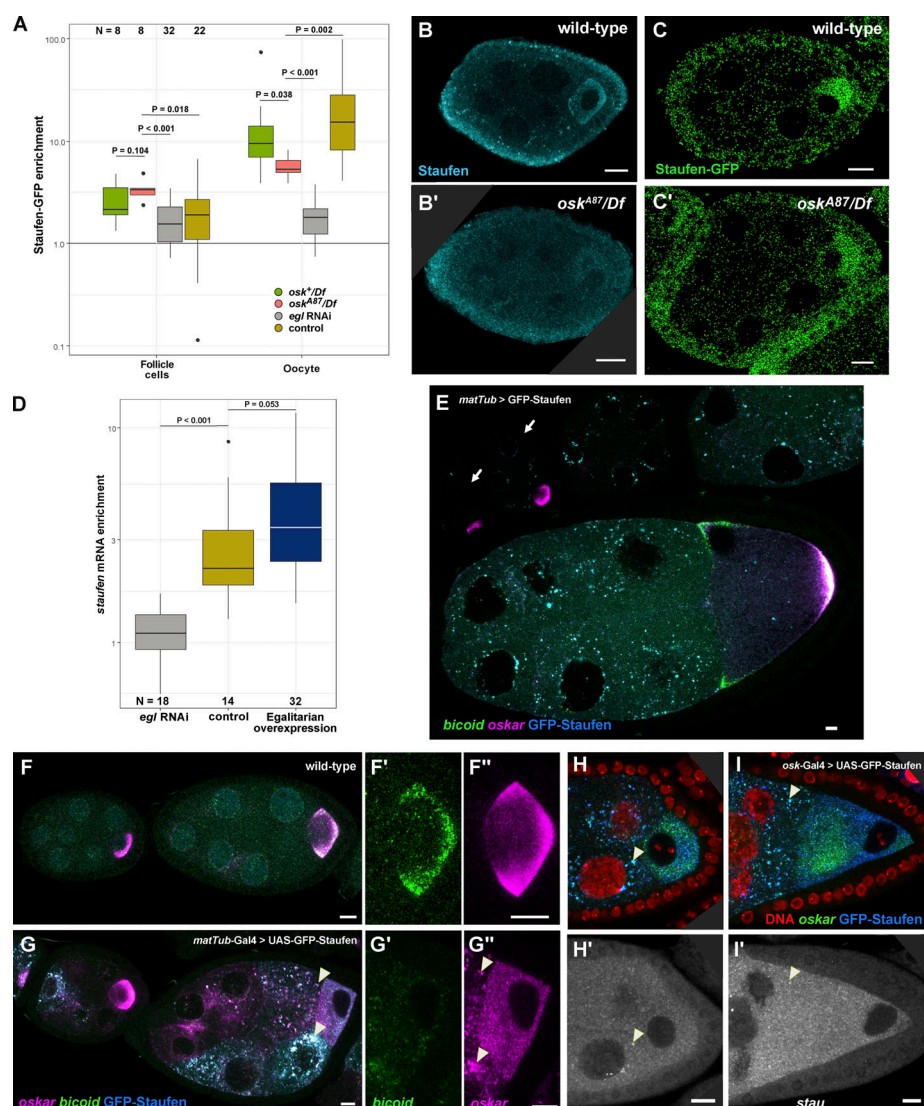

Figure S6. **Localization of Staufen protein and *stau* mRNA in the egg chamber. (A)** Enrichment of Staufen-GFP signal in stage 4–6 oocytes with one (green) or no (red) functional *oskar* alleles expressing *oskar* mRNA, and in oocytes expressing *egl* RNAi (gray) or control RNAi (brown). Enrichment is relative to the sibling nurse cells in the egg chamber. Enrichment of Staufen-GFP in somatic follicle cells, which do not express the shRNA, serves as a control. P values of pairwise Student's *t* test are shown. Note that complete lack of *oskar* mRNA, an abundant binding partner of Staufen, has only a moderate effect on Staufen enrichment in the developing oocyte (also observed in A-B′), whereas knock-down of Egl almost completely abolishes Staufen ooplasmic accumulation. **(B–C′)** Staufen protein expression detected by immunofluorescence (blue, B and B′) or by the fluorescent reporter Staufen-GFP (C and C′, green) in early egg chambers (stage 4–6) in the presence (B and C) and complete absence of *oskar* mRNA (B′ and C′). Note that lack of *oskar* in the oocyte blocks progression of oogenesis beyond stage 6 (Jenny et al., 2006). Scale bars represent 10 μm. **(D)** Quantification of the enrichment of *stau* mRNA in the oocyte in *egl* RNAi (gray), control RNAi (yellow) and Egl overexpressing stage 4–6 egg chambers (dark blue). Note that an excess of Egl has a minuscule effect on *stau* RNA accumulation in the oocyte, suggesting that in the wild type, most of the *stau* mRNA expressed in the nurse cells is transported into the oocyte. **(E)** Expression of GFP-Staufen under the control of maternal tubulin promoter. GFP-Staufen (cyan) can hardly be detected in early egg chambers (highlighted by arrows), and the forming aggregates remain associated with the nurse cell nuclei until mid-oogenesis. **(F–G″)** Early egg chambers overexpressing GFP-Staufen (F, blue) under control of the *matTub*-Gal4 driver. Note that in control oocytes of a similar stage (E–E″), and in oocytes expressing low levels of GFP-Staufen (F, left egg chamber), *oskar* mRNA (E″ and F″, magenta) is enriched. Enrichment of *oskar* and *bicoid* (E′, F′, green) mRNAs is greatly reduced in oocytes expressing high levels of GFP-Staufen (F, right egg chamber). This phenotype is reproducibly observed in >30 egg chambers derived from three separate crosses. **(H–I′)** Egg chambers in early and mid-oogenesis overexpressing GFP-Staufen (blue) under control of the *oskar*-Gal4 driver from the beginning of oogenesis. The vast majority of such egg chambers fail to develop beyond stage 6, likely as a consequence of greatly reduced ooplasmic accumulation of *oskar* mRNA (green), which appears to be trapped in the nurse cells in large aggregates associated with GFP-Staufen (some examples are highlighted by arrowheads in F–H′). Such aggregates of endogenous Staufen are not observed in wild-type egg chambers (E–E″). Similarly, no accumulation of *stau* mRNA in the oocyte is observed in these *oskar*-Gal4>UAS-GFP-Staufen oocytes (G′ and H′), where *stau* mRNA levels are uniformly high in the germline (compare the signal in the follicle cell layer to that in the nurse cells and in the oocyte; see Fig. 7). **(H and H′)** In the oocytes occasionally escaping early developmental arrest, we invariably observed failure in nuclear migration from the posterior to the anterior, reflecting a defect in repolarization of the oocyte microtubule network (Januschke et al., 2006). Consequently, *oskar* mRNA remains in the center of these oocytes, which—although they complete oogenesis—fail to result in viable progeny. These phenotypes are reproducibly observed in >30 egg chambers derived from three separate crosses. **(E–I′)** Scale bars represent 5 μm. (B′ and H–I′) Empty regions created upon rotating the images are shown in dark gray.

Video 1.    **Ex vivo imaging of *oskMS2*-GFP RNPs in extracts of control RNAi stage 9 oocytes by TIRF microscopy.** Images were collected at 20 frames/s with a playback rate of 120 frames/s. *oskar* RNPs are shown in green, microtubule plus ends are labeled by EB1-mCherry (magenta). Scale bar represents 5 µm. Related to Fig. 1.

Video 2.    **Ex vivo imaging of *oskMS2*-GFP RNPs in extracts of *stau* RNAi stage 9 oocytes by TIRF microscopy.** Images were collected at 20 frames/s with a playback rate of 120 frames/s. *oskar* RNPs are shown in green, microtubule plus ends are labeled by EB1-mCherry (magenta). Note the frequent runs of dim *oskar* RNPs in the *stau* RNAi condition. Scale bar represents 5 µm. Related to Fig. 1.

Video 3.    **Live cell imaging of *oskMS2*-GFP RNPs in control RNAi stage 9 oocytes by confocal microscopy.** Images were collected at 3.3 frames/s with a playback rate of 40 frames/s. Anterior (left) and posterior (right) regions of the same oocyte are shown. *oskMS2*-GFP signal is rendered in blue-yellow to allow visual appreciation of dim (blue) and bright (yellow) *oskar* RNPs. Scale bar represents 5 µm. Related to Fig. 1.

Video 4.    **Live cell imaging of *oskMS2*-GFP RNPs in control (Video 3) and *stau* RNAi (Video 4) stage 9 oocytes by confocal microscopy.** Images were collected at 3.3 frames/s with a playback rate of 40 frames/s. Anterior (left) and posterior (right) regions of the same oocyte are shown. *oskMS2*-GFP signal is rendered in blue-yellow to allow visual appreciation of dim (blue) and bright (yellow) *oskar* RNPs. Scale bar represents 5 µm. Related to Fig. 1.

**Provided online is Table S1. Table S1 shows ssDNA oligos used to synthesize smFISH probes targeting the long 3′ UTR of *stau* A/B isoforms or the 5′ extended regions of *stau* A/C isoforms.**

