## [Peer Review File · The Journal of Cell Biology]

An RNA-based feed-forward mechanism ensures motor switching in oskar mRNA transport

Imre Gaspar, Ly Jane Phea, Mark McClintock, Simone Heber, Simon Bullock, and Anne Ephrussi

Corresponding Author(s): Anne Ephrussi, European Molecular Biology Laboratory; Simon Bullock, MRC Laboratory of Molecular Biology; and Imre Gaspar, European Molecular Biology Laboratory

Review Timeline:

Submission Date:	2023-01-27
Editorial Decision:	2023-03-10
Revision Received:	2023-04-01

Monitoring Editor: Cassandra Ori-McKenney

Scientific Editor: Lucia Morgado-Palacin

Transaction Report:

DOI: <https://doi.org/10.1083/jcb.202301113>

Revision 0

Review #1

1. Evidence, reproducibility and clarity:

Evidence, reproducibility and clarity (Required)

Summary:

It is well established that localization of oskar (osk) RNA in the *Drosophila* ovary proceeds in multiple steps. The first step depends upon dynein and results in delivery of osk into the oocyte. The second step involves kinesin-driven transport of osk to the oocyte posterior pole. The manuscript by Gáspár et al brings together several lines of evidence that support an antagonistic relationship with respect to motor binding between two osk-interacting proteins, Egalitarian (Egl) and Staufen (Stau). As staufen RNA and protein accumulate in the oocyte, Egl dissociates from osk, down-regulating dynein and enabling the second stage of osk transport to begin.

Major comments:

In general the experimental results support the conclusions drawn, and the paper includes a strong mix of in vitro and in vivo approaches. Nevertheless I have a few concerns.

(1) In Fig 1D it is apparent that stau KD increases the speed of both plus-end and minus-end runs to a highly significant degree, not just minus-end runs. The stimulating effect of loss of Stau on speed of plus-end runs is not mentioned in the text, and it perhaps muddies the argument that Stau is simply a negative regulator of dynein-dependent minus-end directed transport. This result needs to be explicitly discussed in the text.

(2) I recognize the importance of quantitative imaging to rigorously measure small differences in localization patterns. Nevertheless I find the data in Fig 3 extremely difficult to interpret. Presumably there is standard deviation everywhere there is green signal, but the magenta signal that corresponds to SD is not visible in most places that are green. I suggest adding to Fig 3 a single representative image for each genotype to illustrate each localization pattern, as well as a much clearer explanation of the quantitative imaging data. Perhaps the quantitative images could be moved to a supplemental figure.

Minor comments:

(1) Color/density scales should be added to Figs 1A and S1A, otherwise the yellow/white signal at the posterior could be interpreted as something other than high abundance.

(2) In Fig 4A and 4C, I find it odd to have different halves of images photographed under different intensity settings and would prefer duplicate whole images.

(3)The references to Fig 3G on page 13 should be corrected to Fig 4G.

2. Significance:

Significance (Required)

The paper represents a substantial advance over existing knowledge and it extends our understanding about how RNAs can shuttle between different motor proteins to achieve a localized pattern. However, the Mohr et al 2021 PLoS Genetics paper covers some of the same ground. As that paper has now been published for several months, I believe a revised version of this paper should discuss that other work more prominently, making it apparent where the two studies concur and where this study extends the conclusions of the other one. If there are any contradictions between the two, those should be made explicit as well.

3. How much time do you estimate the authors will need to complete the suggested revisions:

Estimated time to Complete Revisions (Required)

(Decision Recommendation)

Less than 1 month

Review #2

1. Evidence, reproducibility and clarity:

Evidence, reproducibility and clarity (Required)

In this manuscript, Gáspár et al. investigated the molecular mechanisms underlying the switching of motors for *osk* mRNA transport in the *Drosophila* ovary: from dynein in the nurse cells to kinesin-1 in the oocyte. They demonstrated that it requires two RNA-binding proteins, Egalitarian (Egl) and Staufén (Stau) to achieve the posterior localization of *osk* mRNA in the oocyte. Their data show that Egl is responsible for the *stau* mRNA transport into the oocyte, while Stau protein inhibits Egl-dependent dynein transport in the oocyte. Thus, they proposed a feed-forward mechanism in which Egl transports mRNA encoding its own antagonist Stau into the oocyte and thus achieves the switch of the *osk* mRNA transport from dynein to kinesin-1.

The antagonistic interaction between Egl and Staufén is well documented both in vitro and in vivo. All the results are carefully analyzed, but the data presentation is not reader-friendly. Overall, our main concern is about the role of Staufén in *osk* mRNA transport.

****Here are specific points:****

(1) According to the model, lack of Stau should result in failure of displacing Egl from the RNP complex and thus more dynein-driven transport in the oocyte. However, the increase of minus-end run length in stau-RNAi is very small (Figure 1E). It makes us wonder whether Stau is not a dominant inhibitor of Egl/dynein transport of osk RNPs. On the other hand, the speed increase of minus-end run in stau-RNAi is more dramatic than the run length (Figure 1D-1E). Does it mean that in stau-RNAi dynein-driven osk transport has a shorter duration of run? Additionally, in Figure 1D, there is a statistically-significant increase of plus-end-directed transport velocity in stau-RNAi. While the author did mention that in the results "analysis of the speed and length of oskar RNP runs in ooplasmic extracts indicated that Khc activity was not compromised upon staufen knock-down", it does not explain the increased velocity towards the plus-end.

(2) What happened to osk mRNP transport in nurse cells with Staufen overexpression? The authors briefly mentioned that "GFP-Staufen overexpression has no major effect on the localization of oskar (Fig S1F-I)" on page 10. This is quite puzzling, as the authors propose that Staufen antagonized the Egl/dynein-driven transport. If the model holds true, we would expect to see that overexpression of Staufen causes less osk transport in nurse cells and thus less osk accumulated in the oocyte. Can the authors examine the osk mRNP transport in nurse cells in control and in GFP-Staufen overexpressing mutant and quantify the total amount of osk mRNA in the oocyte in control and after GFP-Staufen overexpression?

(3) Is osk mRNP transport in the nurse cells affected by stau-RNAi? The authors showed the Khc association with oskar mRNPs in the nurse cells in Figure 1C. We hope they could quantify the velocity and run length of the osk mRNP particles in nurse cells and compare control with stau-RNAi.

(4) The kymograms of in vitro motility assays (Figure 2A and Figure S2) clearly showed two different moving populations, fast and slow. Did the authors include both types of events in their quantifications? What are the N numbers for each quantification? What do the dots mean in Figure 2B-2G? Does each dot represent a single track in the kymograph? If so, we believe that the sample sizes are too small for in vitro motility assay.

(5) The in vitro motility assay showed that Staufen impairs dynein-driven transport of osk 5'-UTR (Figure 2). Based on these data, it is unclear whether the effect of Staufen is osk mRNA-dependent or Egl-dependent. We suggest performing the motility assay in the absence of osk 5'-UTR and Egl. Dynein, dynactin, and BicD should be sufficient to constitute the processive dynein complex in vitro. The addition of Staufen to the dynein complex will help to understand whether Staufen could directly affect dynein activity. We bring up this point because we noticed that the Staufen displacement of Egl in osk RNPs does not alter the amount of dynein complex associated (Figure 6), implying that Staufen inactivates dynein activity on the RNP complex, independently of Egl-driven dynein recruitment.

(6) In Figure 4, it is hard to see any colocalization between GFP and osk mRNA. And the authors compared overexpressed Egl-GFP (driven by mat atub-Gal4 in mid-oogenesis) with Staufen-

GFP under its endogenous promoter. An endogenous promoter-driven Egl-GFP would be much more appropriate for the comparison.

(7) In a recent publication (Mohr et al., 2021), a different model was proposed, in which Egl mediates transport, and Staufen facilitates the dissociation from the transport machinery for posterior anchoring. Although the authors referred to their paper in the discussion, they should acknowledge the differences and try to reconcile it (at least in the discussion).

(8) In the feed-forward model, Egl is required for the *staufen* mRNA transport from the nurse cells to the oocyte. Are Egl-GFP dots colocalized with *staufen* mRNAs in the nurse cells? Furthermore, to our understanding, in this model, the translation of the *staufen* mRNA would be critical for the switching motors between dynein and kinesin-1. In this sense, *staufen* mRNA translation is either suppressed in the nurse cells or only activated in the oocytes. I think the authors should at least address this point in the discussion.

****Minor points:****

1) I hope the authors would show the *osk* mRNA localization in *egl* mutant in individual stage 9 egg chambers. I can only find the *osk* mRNA in *egl*-RNAi early stage egg chambers (Figure 7E), in which *osk* mRNA still shows an accumulation in the oocyte, although to a much lesser extent compared to control. In another publication (Sanghavi et al., 2016), it seems that the knockdown of Egl by RNAi causes some retention of *osk* mRNA in the nurse cells; but there are still noticeable amount of *osk* mRNA in the oocyte (Figure 3A-B). We wonder whether the authors could quantify the amount of *osk* mRNA both in the nurse cells and in the oocyte of control and *egl*-RNAi. Also I wonder whether the authors could comment on fact that some *osk* mRNA transported into the oocyte. Could it be due to an *egl*-independent transport mechanism?

2) It is always nice to how the average distribution of *osk* mRNA (e.g., Figure 3, Figure S1, and Figure S3). But we recommend having a representative image of each genotype (a single egg) next to the average distribution. It will help the readers to better appreciate the differences among these genotypes.

3) The figure legends are overall hard to read and sometimes impossible to get information about the experiments (for example, Figure 4 legend). Can the authors improve their figure legends making them reader-friendly?

4) For moderate overexpression, the authors used P_{*matα4*-GAL-VP16} (FBtp0009293). However, there are two different transgenic lines associated with FBtp0009293 (V2H and V37), which have slightly different expression levels. The authors should specify which line they used in the experiments.

5) On page 13 "PCR on egg-chambers co-expressing Egl-GFP and either *staufen* RNAi or a control RNAi (white) in the germline (Fig 3G)", it should be Figure 4G.

2. Significance:

Significance (Required)

see above

3. How much time do you estimate the authors will need to complete the suggested revisions:

Estimated time to Complete Revisions (Required)

(Decision Recommendation)

Between 1 and 3 months

Review #3

1. Evidence, reproducibility and clarity:

Evidence, reproducibility and clarity (Required)

Some additional experimental evidence is needed to solidify the conclusions and provide definitive support for this model, as discussed below.

Biochemical experiments using UV crosslinking and GFP immunoprecipitation followed by quantitative PCR were performed to show that Staufen antagonizes the association of Egl with oskar mRNA in vivo. -The authors need to show the quantitative analysis, which was not present in the figure, specifically the effects of Staufen RNAi compared to control.

Is the ability of Staufen to antagonize and displace Egl dependent on Staufen binding to Oscar RNA? Will a Staufen mutant that can't bind to RNA also displace Egl? Alternatively, the mechanism may be independent of RNA binding and perhaps due to protein-protein interactions.

A key question addressed is how does Staufen play a role in directing Oscar RNA localization to the posterior pole. The spatiotemporal control of Staufen at stage 9 seems to be a critical step. A number of experiments are performed to show that Staufen RNA enters the oocyte and accumulates to anterior pole through a process dependent on Egl (Fig. 7).

-Definitive evidence is needed to show the role of 3'UTR of Stau and Egl binding. As it stands now, no evidence is presented to prove that delivery of staufen RNA via Egl, rather than dumping of Staufen protein into oocytes is the necessary trigger for the switch. It is well known that Staufen protein is also transported through ring canals to deliver Staufen into oocytes. There is no need to invoke an additional mechanism of Egl mediated staufen mRNA delivery. A key experiment is to perturb the Egl interaction with staufen 3'UTR and show this is a necessary component to impact oscar. Related to this comment, they should first perform biochemistry IP

and PCR to demonstrate association of Egl with staufer RNA, and then somehow perturb this interaction to assess effects on oscar RNA localization. For example, is the 3'UTR of staufer RNA necessary for this mechanism? What if staufer RNA was ectopically localized in some inappropriate manner, for example localized to posterior pole? Would this prevent the switch of oscar RNA to move to posterior pole? The key question is: is it necessary that translation of Stau be coupled to Egl in order to drive the switch.

****Minor comments****

"Substantially more oscar mRNA was co-immunoprecipitated with Egl-GFP from extracts of egg-chambers expressing staufer RNAi compared to the control (Fig 3G). -This data is not shown in 3G, but rather only in Fig. S4H which needs quantitative analysis shown.

"Addition of recombinant Staufer to the Egl, BicD, dynein and dynactin assembly mix significantly reduced the number of oscar mRNA transport events (Fig. 2A and B)."

-In Fig. 2A, the Y axis shows velocity not number of transport events

Fig. 3. - This is very unclear figure as to what is being shown. More details are needed to explain the figure, and add arrows to help reader note what is being described.

Staufer may also be required for the efficient release of the mRNA from the anterior cortex. This may reflect a role of Staufer in the coupling of the mRNA to the kinesin-dependent posterior transport pathway. This could be discussed as another aspect of the inhibition of dynein and handoff to kinesin.

2. Significance:

Significance (Required)

This elegant manuscript by Gaspar et al provides new insight into the spatiotemporal regulation of Staufer mediated localization of oscar mRNA to the posterior pole in *Drosophila* oocytes. Here the authors demonstrate the competitive displacement of the RNA binding protein Egalitarian, which antagonizes dynein dependent localization at the anterior pole. This work done in this well characterized model of mRNA localization in *Drosophila* oocytes has broader implications for how the bidirectional transport of mRNAs is regulated in other polarized and highly differentiated cells, where very little is known about how mRNA transport direction might be regulated by opposing activities of kinesin and dynein motors. The strengths of this study are the integration of microscopy, biochemistry and genetic mutants to provide very nice experimental support for the two major aspects to the proposed model: 1) the competition between Staufer and Egl on oscar RNA which affects localization, 2) evidence for Egl mediated localization of staufer RNA into the oocyte as a key trigger for competitive displacement to bias localization of oscar RNA via kinesin. However, some additional experimental evidence is needed to solidify the conclusions and provide definitive support for this model, as discussed in other section.

3. How much time do you estimate the authors will need to complete the suggested revisions:

Estimated time to Complete Revisions (Required)

(Decision Recommendation)

Less than 1 month

Manuscript number: 2021-01157

Corresponding author(s): Imre Gaspar, Simon Bullock, Anne Ephrussi

1. General Statements [optional]

We are very grateful to the reviewers for their constructive comments. In response to their critiques, we have made extensive modifications to the manuscript, including documenting new experiments and analyses, and improving data presentation. Here we provide a point-by-point response to the reviewers' comments. We apologize for the delay in resubmitting, which was due to the first (and co-corresponding) author moving to a different position.

Reviewer #1 (Evidence, reproducibility and clarity (Required)):

Summary:

It is well established that localization of oskar (*osk*) RNA in the *Drosophila* ovary proceeds in multiple steps. The first step depends upon dynein and results in delivery of *osk* into the oocyte. The second step involves kinesin-driven transport of *osk* to the oocyte posterior pole. The manuscript by Gáspár et al brings together several lines of evidence that support an antagonistic relationship with respect to motor binding between two *osk*-interacting proteins, Egalitarian (*Egl*) and Staufen (*Stau*). As *stau* RNA and protein accumulate in the oocyte, *Egl* dissociates from *osk*, down-regulating dynein and enabling the second stage of *osk* transport to begin.

Major comments:

In general the experimental results support the conclusions drawn, and the paper includes a strong mix of *in vitro* and *in vivo* approaches. Nevertheless I have a few concerns.

(1) In Fig 1D it is apparent that *stau* KD increases the speed of both plus-end and minus-end runs to a highly significant degree, not just minus-end runs. The stimulating effect of loss of *Stau* on speed of plus-end runs is not mentioned in the text, and it perhaps muddies the argument that *Stau* is simply a negative regulator of dynein-dependent minus-end directed transport. This result needs to be explicitly discussed in the text.

We thank the reviewer for this important comment. Indeed, our previous analysis of the overall population of *oskar* RNPs showed that plus-end-directed runs had increased velocity in the absence of *Staufen* (although the magnitude of the effect was considerably smaller than

observed for minus-end-directed runs). The reviewer's comment prompted us to analyze the effects on motility in more detail. In particular, we have now stratified the data based on the RNA content of the RNPs to control for effects of Staufen depletion on RNA copy number of the motile *oskar* RNPs. These analyses, which are documented in Fig 1B-F of the revised manuscript and discussed between lines 103 and 141 (merged PDF file), indicate that the previous velocity and run length data was somewhat confounded by the Staufen-depleted condition having a lower fraction of moving complexes with a large RNA content, which generally move more slowly. Accounting for this effect shows that impairing Staufen has no significant effect on plus-end-directed run lengths, whereas minus-end-directed run lengths are substantially increased. The velocity of runs is also specifically increased in the minus-end direction in the Staufen-depleted background for RNPs that have a relative RNA content of 1 or 2 units, which represent the majority of the RNP population in that genotype. Whilst RNPs with larger RNA content (2 relative units) do have significantly higher plus-end-directed velocity compared to the same category in the control, the effect is of much smaller magnitude than observed for minus-end-directed movements by this population. To help clarify these results, magnitudes of the effects are now shown in the new Fig. 1 E and F.

These data strengthen the case that Staufen predominantly affects minus-end-directed motion. Given many documented examples of the interdependence of dynein and kinesin on bidirectional cargoes (Hancock et al. 2014), it is conceivable that the modest effects on plus-end-directed velocity for a subset of RNPs arise indirectly from the influence of Staufen on dynein activity. However, we agree with the reviewer that we should not rule out the alternative possibility that Staufen has additional roles in regulating *oskar* transport, including potentially modulating kinesin-1 directly. We have therefore added a section to the Discussion that covers this issue (lines 503-521).

(2) I recognize the importance of quantitative imaging to rigorously measure small differences in localization patterns. Nevertheless I find the data in Fig 3 extremely difficult to interpret. Presumably there is standard deviation everywhere there is green signal, but the magenta signal that corresponds to SD is not visible in most places that are green. I suggest adding to Fig 3 a single representative image for each genotype to illustrate each localization pattern, as well as a much clearer explanation of the quantitative imaging data. Perhaps the quantitative images could be moved to a supplemental figure.

Reviewer 2 also suggested that we include representative images in addition to the quantitative readout. We have now replaced the old Figure 3 with a new one showing representative examples of *oskar* distribution in the different genotypes and moved the quantitative images to the supplement (Figure S4). We have also improved the legends and labeling of this supplementary figure to add clarity.

Minor comments:

Full Revision

(1) Color/density scales should be added to Figs 1A and S1A, otherwise the yellow/white signal at the posterior could be interpreted as something other than high abundance.

We thank the reviewer for spotting this. We have now added a color scale to the relevant figures.

(2) In Fig 4A and 4C, I find it odd to have different halves of images photographed under different intensity settings and would prefer duplicate whole images.

We used this layout to illustrate in the most compact way possible the (co)localization of the two RBPs and *oskar* RNA in the nurse cell and oocyte compartments, where signal intensities can differ dramatically. Following the reviewer's comment, we now show whole images with different intensity settings (Figure 4 A, A', C, C').

(3) The references to Fig 3G on page 13 should be corrected to Fig 4G.

We thank the reviewer for spotting this error, which has now been corrected.

Reviewer #1 (Significance (Required)):

The paper represents a substantial advance over existing knowledge and it extends our understanding about how RNAs can shuttle between different motor proteins to achieve a localized pattern. However, the Mohr et al 2021 PLoS Genetics paper covers some of the same ground. As that paper has now been published for several months, I believe a revised version of this paper should discuss that other work more prominently, making it apparent where the two studies concur and where this study extends the conclusions of the other one. If there are any contradictions between the two, those should be made explicit as well.

We had discussed the Mohr et al. study in our manuscript, which came out when our work was in preparation. Following the reviewer's comment, we now address explicitly how our study differs from theirs and how our work extends their findings. The relevant paragraphs in the Discussion begin on lines 442 and 503. Briefly, a key point of difference is that Mohr et al. focused on the Transport and Anchoring Sequence (TAS) (including its ability to associate with Egl) and other Staufen recognition sites (SRSs) in *oskar* mRNA. Their study also includes an experiment examining the effect of Egl overexpression on *oskar* localization (as described in our original submission). In contrast, our study directly examines the interplay between the RBPs Staufen and Egl on *oskar* RNPs. We are the first to show that Staufen directly antagonizes dynein-based transport and that this is associated, at least in part, with an ability to impair Egl association with RNPs. Moreover, we provide insights into the *in vivo* role of Egl/BicD in recruitment vs activation of dynein on RNPs and how the activity of Staufen is coordinated in space and time via Egl-mediated delivery of *stau* mRNA, which constitutes a novel type of feed-forward mechanism. We do not believe there are any contradictions between the two studies.

Reviewer #2 (Evidence, reproducibility and clarity (Required)):

In this manuscript, Gáspár et al. investigated the molecular mechanisms underlying the switching of motors for *osk* mRNA transport in the *Drosophila* ovary: from dynein in the nurse cells to kinesin-1 in the oocyte. They demonstrated that it requires two RNA-binding proteins, Egalitarian (Egl) and Staufen (Stau) to achieve the posterior localization of *osk* mRNA in the oocyte. Their data show that Egl is responsible for the *stau* mRNA transport into the oocyte, while Stau protein inhibits Egl-dependent dynein transport in the oocyte. Thus, they proposed a feed-forward mechanism in which Egl transports mRNA encoding its own antagonist Stau into the oocyte and thus achieves the switch of the *osk* mRNA transport from dynein to kinesin-1.

The antagonistic interaction between Egl and Staufen is well documented both in vitro and in vivo. All the results are carefully analyzed, but the data presentation is not reader-friendly. Overall, our main concern is about the role of Staufen in *osk* mRNA transport.

****Here are specific points:****

(1) According to the model, lack of Stau should result in failure of displacing Egl from the RNP complex and thus more dynein-driven transport in the oocyte. However, the increase of minus-end run length in *stau*-RNAi is very small (Figure 1E). It makes us wonder whether Stau is not a dominant inhibitor of Egl/dynein transport of *osk* RNPs. On the other hand, the speed increase of minus-end run in *stau*-RNAi is more dramatic than the run length (Figure 1D-1E). Does it mean that in *stau*-RNAi dynein-driven *osk* transport has a shorter duration of run? Additionally, in Figure 1D, there is a statistically-significant increase of plus-end-directed transport velocity in *stau*-RNAi. While the author did mention that in the results "analysis of the speed and length of *oskar* RNP runs in ooplasmic extracts indicated that Khc activity was not compromised upon *stau* knock-down", it does not explain the increased velocity towards the plus-end.

We thank the reviewer for these insightful comments.

We and others (Zimyanin et al. 2008; Gaspar et al., 2014) have shown that there is only a small posterior-directed bias in *oskar* RNP transport in the wild-type ooplasm at mid-oogenesis. Thus, small increases in minus-end-directed transport parameters are expected to be sufficient for anterior mislocalization of a subset of RNPs, as is seen in *stau* mutants (note that we would not expect a dramatic increase in minus-end-directed motile properties in the *stau* RNAi condition, as a significant fraction of *oskar* RNA is targeted posteriorly). To allow the readers to better judge the magnitude of the effects, we now include the percentage change in mean velocity and run length values on the graphs (new Figure 1E and F).

Regarding the reviewer's question about the run duration, indeed it is shorter for the minus-end directed runs in the absence of Staufen. In the motor field, it is typical to present velocity and run length only because duration is dependent on these two parameters.

Reviewer 1 also made a similar comment about plus-end directed velocity of RNPs. As we wrote in response to their comment, we have now stratified the data based on the RNA content of the RNPs to control for effects of Staufen depletion on RNA copy number of the motile *oskar* RNPs. These analyses, which are documented in Fig 1 B-F of the revised manuscript and discussed between lines 103 and 141, indicate that the previous velocity and run length data were somewhat confounded by the Staufen-depleted condition having a lower fraction of moving complexes with a large RNA content, which generally move more slowly. Accounting for this effect shows that impairing Staufen has no significant effect on plus-end-directed run lengths, whereas minus-end-directed run lengths are substantially increased. The velocity of runs is also increased only in the minus-end direction in the Staufen-depleted background for RNPs that have a RNA content of 1 or 2 relative units, which represent the majority of the RNP population in that genotype. Whilst RNPs with larger RNA content (2 relative units) do have significantly higher plus-end-directed velocity compared to the same category in the control, the effect is of much smaller magnitude than observed for minus-end-directed movement for this population.

These data strengthen the case that Staufen predominantly affects minus-end-directed motion. Given many documented examples of the interdependence of dynein and kinesin on cargoes (Hancock et al., 2014), it is conceivable that the modest effects on plus-end-directed velocity arise indirectly due to the influence of Staufen on dynein activity. However, we agree with the reviewer that we should not rule out the alternative possibility that Staufen has additional roles in regulating *oskar* transport, including potentially modulating kinesin-1 activity directly. We have therefore added a section to the Discussion that covers this issue (lines 503-521).

(2) What happened to *osk* mRNP transport in nurse cells with Staufen overexpression? The authors briefly mentioned that "GFP-Staufen overexpression has no major effect on the localization of *oskar* (Fig S1F-I)" on page 10. This is quite puzzling, as the authors propose that Staufen antagonized the Egl/dynein-driven transport. If the model holds true, we would expect to see that overexpression of Staufen causes less *osk* transport in nurse cells and thus less *osk* accumulated in the oocyte. Can the authors examine the *osk* mRNP transport in nurse cells in control and in GFP-Staufen overexpressing mutant and quantify the total amount of *osk* mRNA in the oocyte in control and after GFP-Staufen overexpression?

We showed in the initial submission that strong overexpression of GFP-Staufen in early oogenesis (e.g. with *osk*-Gal4) disrupts *oskar* localization, including causing ectopic accumulation in the nurse cells (Fig S7F and G, now marked with arrowheads). Fig S1F-I, to which the reviewer refers, documents an experiment in which the expression of GFP-Staufen was directly driven by the maternal tubulin promoter (i.e. not through the UAS-Gal4 system; now indicated in Fig. S1F). We had assumed that the difference in behavior of the different GFP-

Staufen transgenes was caused by the timing and the amount of overexpression – maternal Gal4 drivers are capable of very strong and, in the case of *osk-Gal4*, early expression of UAS transgenes. Prompted by the reviewer, we have now examined GFP-Staufen expression in these lines in more detail. This confirmed our previous assumptions about timing and levels of ectopic expression. We now included a new panel Fig S7I to document the expression of maternal tubulin promoter-driven GFP-Staufen and have updated the manuscript to include details about the mode of Staufen overexpression used in different experiments (lines 205, 411-415).

(3)Is *osk* mRNP transport in the nurse cells affected by *stau*-RNAi? The authors showed the Khc association with *oskar* mRNPs in the nurse cells in Figure 1C. We hope they could quantify the velocity and run length of the *osk* mRNP particles in nurse cells and compare control with *stau*-RNAi.

We have never succeeded in making squashes of nurse cells that maintain *osk*^{MS2} RNA transport. Therefore, we are unable to evaluate directional transport of *oskar* in these cells. However, Staufen does not accumulate to appreciable levels in the nurse cells, as shown by Little et al., 2015 and also Figure 4A and A' (left panels). Moreover, we did not detect significant colocalization between Staufen and *oskar* in the nurse cells (Fig. 4B). Therefore, depletion of Staufen with RNAi is not expected to influence motility of *oskar* in this part of the egg chamber.

(4)The kymograms of in vitro motility assays (Figure 2A and Figure S2) clearly showed two different moving populations, fast and slow. Did the authors include both types of events in their quantifications? What are the N numbers for each quantification? What do the dots mean in Figure 2B-2G? Does each dot represent a single track in the kymograph? If so, we believe that the sample sizes are too small for in vitro motility assay.

For completeness, we did not exclude particles from our analysis based on their speed of movement. We have now made this point clear in an updated section of the Methods (lines 807-811), which provides additional information on particle inclusion criteria.

We did document in the legends what the dots represent (values for single microtubules). We have now also included information on the number of complexes analyzed, which is 586-1341 single RNA particles or 1247-2207 single dynein particles per condition. These sample sizes are considerably larger than those used in most *in vitro* motility studies.

(5)The in vitro motility assay showed that Staufen impairs dynein-driven transport of *osk* 5'-UTR (Figure 2). Based on these data, it is unclear whether the effect of Staufen is *osk* mRNA-dependent or Egl-dependent. We suggest performing the motility assay in the absence of *osk* 5'-UTR and Egl. Dynein, dynactin, and BicD should be sufficient to constitute the processive dynein complex in vitro. The addition of Staufen to the dynein complex will help to understand whether Staufen could directly affect dynein activity. We bring up this point because we noticed that the Staufen displacement of Egl in *osk* RNPs does not alter the amount of dynein complex

associated (Figure 6), implying that Staufen inactivates dynein activity on the RNP complex, independently of Egl-driven dynein recruitment.

We cannot look at transport of dynein in the presence of only dynactin and full-length BicD as BicD is not activated (and thus unable to effectively bind dynein and dynactin) without Egl and RNA (McClintock et al. 2018, Sladewski et al. 2018). However, the reviewer's comment prompted us to investigate the effect of Staufen on dynein-dynactin motility that is stimulated by the constitutively active truncated mammalian BicD2, so called BicD2N (Schlager et al. 2014, McKenney et al. 2014). We find that Staufen partially inhibits DDB motility but not to the extent seen with the full-length BicD in the presence of Egl and RNA (new main figure panels 2H and I, and Figure S3). As stated on lines 186 - 188, these data suggest that Staufen inhibits both the activation of dynein-dynactin motility by BicD proteins, as well as stimulation of this event by Egl and RNA. This finding is also incorporated in a new section of the Discussion that covers possible roles of Staufen in addition to competing for Egl's binding to RNA (between lines 503-521). We are very grateful to the reviewer for suggesting this approach, as it has provided significant new insight into Staufen's function.

(6) In Figure 4, it is hard to see any colocalization between GFP and *osk* mRNA. And the authors compared overexpressed Egl-GFP (driven by *mat atub-Gal4* in mid-oogenesis) with Staufen-GFP under its endogenous promoter. An endogenous promoter-driven Egl-GFP would be much more appropriate for the comparison.

Colocalization between GFP and *oskar* signals is seen as white in Fig. 4A and C. We have now added arrows to highlight a few examples of colocalization. The degree of colocalization was quantified in an unbiased fashion (shown in panels Fig 4B and D).

Regarding the expression of Egl-GFP: it was driven directly by the *aTub84B* promoter and not by *matTub-Gal4*. Western blot analysis performed in response to the reviewer's comment shows that Egl-GFP is expressed at similar levels to endogenous Egl in this line (new Fig. S5I).

(7) In a recent publication (Mohr et al., 2021), a different model was proposed, in which Egl mediates transport, and Staufen facilitates the dissociation from the transport machinery for posterior anchoring. Although the authors referred to their paper in the discussion, they should acknowledge the differences and try to reconcile it (at least in the discussion).

We now further discuss our work in the light of the findings by Mohr et al. (a request also made by Reviewer 1) (in paragraphs starting on lines 442 and 503). In our opinion, the data of Mohr et al. in fixed material cannot discriminate between effects of Staufen (or the TAS) on transport vs anchorage. In contrast, our dynamic imaging *in vitro* and *ex vivo* shows unambiguously that Staufen can modulate transport processes. As accumulation of RNA at the cortex is dependent on directional transport, we do not think it necessary to invoke a separate anchorage role of Staufen. We have now raised the possibility that transport and cortical localization are two

facets of the same underlying process in the hope that this will stimulate further investigation (lines 462-466).

(8) In the feed-forward model, Egl is required for the *stau* mRNA transport from the nurse cells to the oocyte. Are Egl-GFP dots colocalized with *stau* mRNAs in the nurse cells?

We showed in Fig 7I of the original submission that Egl-GFP puncta are colocalized with *stau* mRNAs in nurse cells. Indeed, this is a key piece of evidence for our model. These data are now in Figure 7F.

Furthermore, to our understanding, in this model, the translation of the *stau* mRNA would be critical for the switching motors between dynein and kinesin-1. In this sense, *stau* mRNA translation is either suppressed in the nurse cells or only activated in the oocytes. I think the authors should at least address this point in the discussion.

This is another excellent suggestion. We have now included in the Discussion (from line 533) the point that *Staufen* translation may be suppressed during transit to the oocyte or that the protein may be translated en route but only build up to meaningful levels where the RNA is concentrated in the oocyte.

Minor points:

1) I hope the authors would show the *osk* mRNA localization in *egl* mutant in individual stage 9 egg chambers. I can only find the *osk* mRNA in *egl*-RNAi early stage egg chambers (Figure 7E), in which *osk* mRNA still shows an accumulation in the oocyte, although to a much lesser extent compared to control. In another publication (Sanghavi et al., 2016), it seems that the knockdown of Egl by RNAi causes some retention of *osk* mRNA in the nurse cells; but there are still noticeable amount of *osk* mRNA in the oocyte (Figure 3A-B). We wonder whether the authors could quantify the amount of *osk* mRNA both in the nurse cells and in the oocyte of control and *egl*-RNAi. Also I wonder whether the authors could comment on fact that some *osk* mRNA transported into the oocyte. Could it be due to an *egl*-independent transport mechanism?

egl null mutants do not reach stage 9 due to a defect in retention of oocyte fate, hence the use of *egl* RNAi in our study and the one by Sanghavi et al. Whilst we can't rule out a (minor) Egl-independent mechanism for localizing *oskar* RNA in the oocyte, to date no other pathway has been implicated in the delivery of this or any other mRNA from the nurse cells. We favor a scenario in which residual *oskar* accumulation in the oocyte in *egl* RNAi egg chambers is due to incomplete depletion of Egl protein in the knockdown condition. We have noted this in the relevant figure legend and also clarify that the RNAi is a tool for knockdown in line 396 of the Results section.

The below plot shows a quantification of *oskar* mRNA localization in *egl* and control RNAi egg chambers, which the reviewer was wondering about.

In the *egl* RNAi egg-chambers, there is a significant increase in the mean signal intensity of *oskar* mRNA in the nurse cells, while *oskar* mRNA levels are substantially reduced in the oocyte, in line with the findings of Sanghavi et al., 2016.

2) It is always nice to have the average distribution of *osk* mRNA (e.g., Figure 3, Figure S1, and Figure S3). But we recommend having a representative image of each genotype (a single egg) next to the average distribution. It will help the readers to better appreciate the differences among these genotypes.

This suggestion was also made by Reviewer 1. We have added representative images to Figure 3 and moved the images depicting average distributions to the supplement (Fig S4). We have also improved the legend and labeling for Fig S4.

3) The figure legends are overall hard to read and sometimes impossible to get information about the experiments (for example, Figure 4 legend). Can the authors improve their figure legends making them reader-friendly?

We have edited the legends to make them clearer, including an extensive reworking of those for Figure 4. We thank the reviewer for encouraging us to do this.

4) For moderate overexpression, the authors used P{mata4-GAL-VP16} (FBtp0009293). However, there are two different transgenic lines associated with FBtp0009293 (V2H and V37), which have slightly different expression levels. The authors should specify which line they used in the experiments.

The matTub-Gal4 transgene we used in our study is inserted in the 2nd chromosome. We now mention this in the Methods section (line 574). We received this line from another lab many years ago, with no additional information provided.

5) On page 13 "PCR on egg-chambers co-expressing Egl-GFP and either staufer RNAi or a control RNAi (white) in the germline (Fig 3G)", it should be Figure 4G.

We apologize for this mistake, which has now been fixed.

Reviewer #2 (Significance (Required)):

see above

Reviewer #3 (Evidence, reproducibility and clarity (Required)):

Some additional experimental evidence is needed to solidify the conclusions and provide definitive support for this model, as discussed below.

Biochemical experiments using UV crosslinking and GFP immunoprecipitation followed by quantitative PCR were performed to show that Staufer antagonizes the association of Egl with oskar mRNA in vivo. -The authors need to show the quantitative analysis, which was not present in the figure, specifically the effects of Staufer RNAi compared to control.

These quantitative data, which are key for our model, were shown in the original submission (Fig 4G in the original and revised manuscript). We mistakenly called out the panel as 3G in the original submission. We apologize for this error, which has now been dealt with.

Full Revision

Is the ability of Staufen to antagonize and displace Egl dependent on Staufen binding to Oscar RNA? Will a Staufen mutant that can't bind to RNA also displace Egl? Alternatively, the mechanism may be independent of RNA binding and perhaps due to protein-protein interactions.

While the details of how Staufen displaces Egl are certainly an interesting topic for future research, we consider that addressing this goes well beyond the scope of this study, which already covers a lot of ground. Staufen contains four double stranded RNA-binding domains, and deleting or mutating all of these domains is likely to interfere with overall folding of Staufen, thus confounding the interpretation of the results.

As an alternative approach to elucidating RNA-dependent vs RNA-independent roles of Staufen, we have now assessed the effect of the protein on *in vitro* motility of dynein-dynactin complexes formed in the presence of a constitutively active truncation of mammalian BicD2 (BicD2N). We find that Staufen partially inhibits motility of these 'DDB' complexes but not to the extent seen with the full length BicD in the presence of Egl and RNA (new Fig 2H, I and S3). As stated in the manuscript (lines 186-188) these data suggest that Staufen inhibits both the activation of dynein-dynactin motility by BicD proteins, as well as stimulation of this event by Egl and RNA. We believe these experiments provide significant new insight into Staufen's function. This finding is also incorporated into a new section of the Discussion dealing with potential roles of Staufen in addition to displacing Egl from RNPs.

A key question addressed is how does Staufen play a role in directing Oscar RNA localization to the posterior pole. The spatiotemporal control of Staufen at stage 9 seems to be a critical step. A number of experiments are performed to show that Staufen RNA enters the oocyte and accumulates to anterior pole through a process dependent on Egl (Fig. 7).

-Definitive evidence is needed to show the role of 3'UTR of Stau and Egl binding. As it stands now, no evidence is presented to prove that delivery of staufen RNA via Egl, rather than dumping of Staufen protein into oocytes is the necessary trigger for the switch. It is well known that Staufen protein is also transported through ring canals to deliver Staufen into oocytes. There is no need to invoke an additional mechanism of Egl mediated staufen mRNA delivery. A key experiment is to perturb the Egl interaction with staufen 3'UTR and show this is a necessary component to impact oscar. Related to this comment, they should first perform biochemistry IP and PCR to demonstrate association of Egl with staufen RNA, and then somehow perturb this interaction to assess effects on oscar RNA localization. For example, is the 3'UTR of staufen RNA necessary for this mechanism? What if staufen RNA was ectopically localized in some inappropriate manner, for example localized to posterior pole? Would this prevent the switch of oscar RNA to move to posterior pole? The key question is: is it necessary that translation of Stau be coupled to Egl in order to drive the switch.

Mapping of the Egl-binding site in *stau* mRNA is a major undertaking requiring the production and evaluation of multiple new transgenic fly lines. We feel that this would constitute an entirely new study. Moreover, multiple lines of evidence already support a functional interaction between

Egl and *stau* mRNA, notably the presence of Egl on *stau* RNPs (previously Fig. 7I, now Fig. 7F), the strongly impaired accumulation of *stau* mRNA in the oocyte of *egl* RNAi egg chambers, and the ability of Egl overexpression to reposition a subset of the *stau* mRNA population at the anterior cortex.

We have now performed new experiments and analyses to test the alternative hypothesis that Staufen protein is transported into the oocyte in the absence of *stau* mRNA transport. We find that disrupting Egl function with RNAi impairs localisation of both *stau* mRNA and protein in the proto-oocyte (new Figure 7A-D). As Egl has no known function in protein transport, these data argue against an RNA-independent mechanism for Staufen protein delivery. Moreover, we showed that both *stau* mRNA and Staufen are enriched in early oocytes lacking *oskar* mRNA, the main target of Staufen protein in the female germline. This result shows that Staufen protein is not appreciably transported from the nurse cells to the oocyte by hitchhiking on its RNA targets.

Whilst Mhlanga et al. 2009 did report transport of large GFP-Staufen particles through ring canals, the line used (*matTub4>GFP-Staufen* from the St Johnston lab, which was also used for our rescue experiments) is known to make protein aggregates which is not the case for the endogenous protein (Zimyanin et al., 2008 and our new Figures 7B and S7E-I) and are therefore likely to be artefactual. Neither we, nor previous studies (Little et al., 2015), detected endogenous Staufen protein in nurse cells.

Finally, the reviewer asks if coupling Staufen translation to Egl-mediated enrichment of *stau* mRNA in the oocyte is important: we showed in the original submission that strong overexpression of GFP-Staufen by Gal4 drivers leads to mislocalization of Staufen in the nurse cells of early egg-chambers, presumably due to saturation of the Egl-based transport machinery. In these egg-chambers, we observed defects in RNA enrichment in the primordial oocyte and defects in oogenesis, consistent with the need to exclude Staufen protein from the nurse cells.

These findings are now presented in new panels of the updated Figures 7 and S7, with the corresponding section of the manuscript revised accordingly (lines 410-420). We think that altogether these lines of evidence strongly support our model that Egl transports *stau* mRNA into the developing oocyte and that this process is pivotal for *oskar* RNA localization.

****Minor comments****

"Substantially more *oskar* mRNA was co-immunoprecipitated with Egl-GFP from extracts of egg-chambers expressing *stau* RNAi compared to the control (Fig 3G). -This data is not shown in 3G, but rather only in Fig. S4H which needs quantitative analysis shown.

Full Revision

This point stems from us calling out the wrong panel in the first submission; this has now been addressed, as described above. We apologize for the error.

"Addition of recombinant Staufen to the Egl, BicD, dynein and dynactin assembly mix significantly reduced the number of oskar mRNA transport events (Fig. 2A and B)."

-In Fig. 2A, the Y axis shows velocity not number of transport events

Fig 2A is a kymograph that is representative of the overall effect, where the Y-axis represents time. The reviewer may be referring to Fig 2B but this shows the frequency of processive *oskar* RNA movements (expressed as 'number / micron / minute'), not velocity (micron/minute).

Fig. 3. - This is very unclear figure as to what is being shown. More details are needed to explain the figure, and add arrows to help reader note what is being described.

We have changed this figure to show representative images of individual egg chambers, as requested by the other two reviewers. The original Fig 3 is now moved to the Supplement as Fig S4. We have added arrows to the figure to indicate the anterior mislocalization of *oskar* mRNA and edited the legend for clarity.

Staufen may also be required for the efficient release of the mRNA from the anterior cortex. This may reflect a role of Staufen in the coupling of the mRNA to the kinesin-dependent posterior transport pathway. This could be discussed as another aspect of the inhibition of dynein and handoff to kinesin.

This is an interesting idea but it does not fit with our observation that Staufen depletion does not alter the association of *oskar* RNPs with kinesin-1 (originally Fig. 1C, now Fig. 1D). We do, however, now include in the Discussion a section on other ways, in addition to promoting Egl disassociation, that Staufen might orchestrate *oskar* mRNA transport.

Reviewer #3 (Significance (Required)):

This elegant manuscript by Gaspar et al provides new insight into the spatiotemporal regulation of Staufen mediated localization of oscar mRNA to the posterior pole in *Drosophila* oocytes. Here the authors demonstrate the competitive displacement of the RNA binding protein Egalitarian, which antagonizes dynein dependent localization at the anterior pole. This work done in this well characterized model of mRNA localization in *Drosophila* oocytes has broader implications for how the bidirectional transport of mRNAs is regulated in other polarized and highly differentiated cells, where very little is known about how mRNA transport direction might be regulated by opposing activities of kinesin and dynein motors. The strengths of this study are the integration of microscopy, biochemistry and genetic mutants to provide very nice

Full Revision

experimental support for the two major aspects to the proposed model: 1) the competition between Staufen and Egl on oscar RNA which affects localization, 2) evidence for Egl mediated localization of staufen RNA into the oocyte as a key trigger for competitive displacement to bias localization of oscar RNA via kinesin. However, some additional experimental evidence is needed to solidify the conclusions and provide definitive support for this model, as discussed in other section.

March 10, 2023

RE: JCB Manuscript #202301113T

Dr. Anne Ephrussi
European Molecular Biology Laboratory
Meyerhofstrasse 1
Heidelberg 69117
Germany

Dear Dr. Ephrussi:

Thank you for submitting your revised manuscript entitled "An RNA-based feed-forward mechanism ensures motor switching in oskar mRNA transport". The original reviewers have now assessed your revised manuscript and, as you can see, they are satisfied with the revisions. However, one reviewer did note that additional information in the legends would make the paper more streamlined and general copyediting would ensure the correct figures are referenced in the text. We would be happy to publish your paper in JCB pending final revisions to address this minor reviewer points. In your final revision, please ensure that you comply with our formatting guidelines (see details below).

To avoid unnecessary delays in the acceptance and publication of your paper, please read the following information carefully. Please go through all the formatting points paying special attention to those marked with asterisks.

A. MANUSCRIPT ORGANIZATION AND FORMATTING:

1) Text limits: Character count for Articles and Tools is < 40,000, not including spaces. Count includes title page, abstract, introduction, results, and acknowledgments. Count does not include materials and methods, figure legends, references, tables, or supplemental legends.

2) Figures limits: Articles and Tools may have up to 10 main text figures.

***** Please note that main text figures should be provided as individual, editable files.**

3) Figure formatting:

Molecular weight or nucleic acid size markers must be included on all gel electrophoresis.

Scale bars must be present on all microscopy images, including inset magnifications.

Also, please avoid pairing red and green for images and graphs to ensure legibility for color-blind readers. If red and green are paired for images, please ensure that the particular red and green hues used in micrographs are distinctive with any of the colorblind types. If not, please modify colors accordingly or provide separate images of the individual channels.

4) Statistical analysis:

***** Error bars on graphic representations of numerical data must be clearly described in the figure legend.**

***** The number of independent data points (n) represented in a graph must be indicated in the legend. Please, indicate whether 'n' refers to technical or biological replicates (i.e. number of analyzed cells, samples or animals, number of independent experiments).**

If independent experiments with multiple biological replicates have been performed, we recommend using distribution-reproducibility SuperPlots (please, see Lord et al., JCB 2020) to better display the distribution of the entire dataset, and report statistics (such as means, error bars, and P values) that address the reproducibility of the findings.

***** Statistical methods should be explained in full in the materials and methods in a separate section. Please, provide some**

details of the statistical tests you have used in this section.

For figures presenting pooled data the statistical measure should be defined in the figure legends.

Please also be sure to indicate the statistical tests used in each of your experiments (both in the figure legend itself and in a separate methods section) as well as the parameters of the test (for example, if you ran a t-test, please indicate if it was one- or two-sided, etc.).

*** As you used parametric tests in your study (i.e. t-tests), you should have first determined whether the data was normally distributed before selecting that test. In the stats section of the methods, please indicate how you tested for normality. If you did not test for normality, you must state something to the effect that "Data distribution was assumed to be normal but this was not formally tested."

5) Abstract and title:

The abstract should be no longer than 160 words and should communicate the significance of the paper for a general audience.

The title should be less than 100 characters including spaces. Make the title concise but accessible to a general readership.

6) Materials and methods:

Should be comprehensive and not simply reference a previous publication for details on how an experiment was performed. The text should not refer to methods "...as previously described."

Also, the materials and methods should be included in the main manuscript text and not in the supplementary materials.

7) For all cell lines, vectors, constructs/cDNAs, etc. - all genetic material: please include database / vendor ID (e.g., Addgene, ATCC, etc.) or if unavailable, please briefly describe their basic genetic features, even if described in other published work or gifted to you by other investigators (and provide references where appropriate).

Please be sure to provide the sequences for all of your oligos: primers, si/shRNA, RNAi, gRNAs, etc. in the materials and methods.

You must also indicate in the methods the source, species, and catalog numbers/vendor identifiers (where appropriate) for all of your antibodies, including secondary, and the system used to collect the signal from the antibodies. If antibodies are not commercial, please add a reference citation if possible.

8) Microscope image acquisition:

The following information must be provided about the acquisition and processing of images:

- a. Make and model of microscope
- b. Type, magnification, and numerical aperture of the objective lenses
- c. Temperature
- d. imaging medium
- e. Fluorochromes
- f. Camera make and model
- g. Acquisition software
- h. Any software used for image processing subsequent to data acquisition. Please include details and types of operations involved (e.g., type of deconvolution, 3D reconstitutions, surface or volume rendering, gamma adjustments, etc.).

10) Supplemental materials:

*** There are strict limits on the allowable amount of supplemental data. Articles and Tools may have up to 5 supplemental figures. There is no limit for supplemental tables. You currently have 7 supplemental figures. We can give you a bit more space, but we would need you to try to reduce the number of supplementary figures (up to 6 if possible) by consolidating data from two figures into one or moving supplemental data to one of the main figures. Please be sure to correct the callouts in the text to reflect this change.

*** Please note that supplemental figures and tables should be provided as individual, editable files.

*** A summary of all supplemental material should appear at the end of the Materials and Methods section (please see any recent JCB paper for an example of this summary).

11) Video legends:

*** Video legends should describe what is being shown, the cell type or tissue being viewed (including relevant cell treatments, concentration and duration, or transfection), the imaging method (e.g., time-lapse epifluorescence microscopy), what each color represents, how often frames were collected, the frames/second display rate, and the number of any figure that has related video stills or images.

12) eTOC summary:

*** A ~40-50 word summary that describes the context and significance of the findings for a general readership should be included on the title page. The statement should be written in the present tense and refer to the work in the third person. It should begin with "First author name(s) et al..." to match our preferred style.

13) Conflict of interest statement:

JCB requires inclusion of a statement in the acknowledgements regarding competing financial interests. If no competing financial interests exist, please include the following statement: "The authors declare no competing financial interests."

14) Author contribution:

A separate author contribution section is required following the Acknowledgments in all research manuscripts.

*** All authors should be mentioned and designated by their first and middle initials and full surnames and the CRediT nomenclature is encouraged (<https://casrai.org/credit/>).

15) ORCID IDs: ORCID IDs are unique identifiers allowing researchers to create a record of their various scholarly contributions in a single place. At resubmission of your final files, please consider providing an ORCID ID for as many contributing authors as possible.

16) Materials and data sharing:

All animal and human studies must be conducted in compliance with relevant local guidelines, such as the US Department of Health and Human Services Guide for the Care and Use of Laboratory Animals or MRC guidelines, and must be approved by the authors' Institutional Review Board(s). A statement to this effect with the name of the approving IRB(s) must be included in the Materials and Methods section.

*** Journal of Cell Biology now requires a data availability statement for all research article submissions. These statements will be published in the article directly above the Acknowledgments. The statement should address all data underlying the research presented in the manuscript. Please visit the JCB instructions for authors for guidelines and examples of statements at (<https://rupress.org/jcb/pages/editorial-policies#data-availability-statement>).

All datasets included in the manuscript must be available from the date of online publication, and the source code for all custom computational methods, apart from commercial software programs, must be made available either in a publicly available database or as supplemental materials hosted on the journal website. Numerous resources exist for data storage and sharing (see Data Deposition: <https://rupress.org/jcb/pages/data-deposition>), and you should choose the most appropriate venue based on your data type and/or community standard. If no appropriate specific database exists, please deposit your data to an appropriate publicly available database.

17) Please note that JCB now requires authors to submit Source Data used to generate figures containing gels and Western blots with all revised manuscripts. This Source Data consists of fully uncropped and unprocessed images for each gel/blot displayed in the main and supplemental figures. The Source Data files will be directly linked to specific figures in the published article.

*** As your paper includes cropped gel and/or blot images, please be sure to provide one Source Data file for each figure that contains gels and/or blots along with your revised manuscript files. File names for Source Data figures should be alphanumeric without any spaces or special characters (i.e., SourceDataF#, where F# refers to the associated main figure number or SourceDataFS# for those associated with Supplementary figures). The lanes of the gels/blots should be labeled as they are in

the associated figure, the place where cropping was applied should be marked (with a box), and molecular weight/size standards should be labeled wherever possible.

B. FINAL FILES:

Thank you for this interesting contribution, we look forward to publishing your paper in Journal of Cell Biology.

Sincerely,

Kassandra Ori-McKenney
Monitoring Editor
Journal of Cell Biology

Lucia Morgado-Palacin, PhD
Scientific Editor
Journal of Cell Biology

Reviewer #1 (Comments to the Authors (Required)):

The authors added a significant amount of data and new analysis to support their conclusion. I do still have some remaining concerns, mostly regarding their presentation that are easy to fix.

(1) Overall, figures and supplementary figures should be referred to within the main text in numerical order. The authors jumped

between the supplementary figures within the text, which is not reader-friendly. For example, at the beginning of the results: Line 99-101 "However, loss of Staufen does not seem to affect the overall RNA content of oskar RNPs in the oocyte as revealed by single-molecule fluorescent in situ hybridization (smFISH) (Fig S5B)". That is the first time they refer to any supplementary figure in the results, and it is Figure S5. There are numerous similar examples in the text.

(2) In several cases, I was not able to locate panels that were mentioned in the main text. For example, in lines 210-212, the authors said "Further supporting this notion, removal of a functional copy of *egl* partially suppressed oskar mislocalization at the anterior of egg-chambers with reduced Staufen levels (*stau* RNAi, *egl*^{1/+}; Fig S3C-E)". I was not able to locate the Figure S3C-E. Line 364-366: the authors write "However when we analyzed GFP-Dhc association with oskar RNPs, we found no major difference between *stau* RNAi, *Egl*-overexpressing, and control oocytes (Fig 6F and F)". Again, there is no Figure 6F'. Line 367-368: "Intriguingly, *stau* RNAi and *Egl* overexpressing oocytes also did not display an increase in the association of BicD with oskar RNPs (Fig 6G and G)". I cannot locate Figure 6G'. And Figure 6G is not about BicD association either. So where are the data about BicD association with *osk* RNPs?

(3) Figure legends need more information. I don't understand Figure S1I. What are the blue, red, and green arrowheads? Is the GFP-tagged Staufen larger than the endogenous Staufen protein? What are the estimated kDa for Staufen and GFP-Staufen? Does the lane 3 Staufen-GFP refer to the one under the control of its endogenous *stau* promoter? Is the lane 4 GFP-Staufen overexpression referring to UASp-GFP-*stau* under *mat atub-Gal4* driver? All the information should be included in the figure legends. How the fraction in Figure 4F can be negative. Is it normalized to GFP-only control? Please include the related information in the legends. Another example is Figure 6A-C: how are BicD and p50 labeled? By UASp-BicD.eGFP and UAS-DCTN2-p50::GFP, respectively? If so, how are these transgenes driven? Why does *osk* mRNA have some mislocalization phenotype, especially in Figures 6A and 6B? All the information should be included in the Figure legends.

(4) Some figures have the error bars overlapping, making it really hard to distinguish, such as Figure 4E, Figure S5F, and Figure 6D-F. For Figure 4E, is there a statistically significant difference between control (blue) and Bic1/+ (purple)?

Some minor scientific questions:

1. Line 391-392, the authors claimed that "both Staufen and Staufen-GFP signal accumulated in the oocyte even in the absence of oskar mRNA (Fig S7A-C)". However, the endogenous Staufen level is much lower in the oskar mutant background (Figure S7A') compared to the control (Figure S7A). It is not consistent with the text. And I cannot find any quantification of this set of data.

2. What are the egg chamber stages quantified in Figure 7D and Figure S7D? Does the *stau* mRNA oocyte accumulation become less during mid-oogenesis (compared Figure 7E with Figure 7B)? If so, any quantification and explanation?

3. What MCP-GFP line was used for the live imaging? I cannot find any information about the line: which promoter and which chromosome insertion? No reference either.

4. Line 497-498: "dynein's association with oskar RNPs in the oocyte does not scale with RNA content". This is slightly misleading, as the authors showed that the Dhc amount does increase with the numbers of RNA per RNP (Figure 6E).

5. In Figure 1, the *osk*-MS2 motility was analyzed in ooplasmic extracts. How was the movement direction determined? According to the Methods, these samples were labeled with mCherry- α -tubulin. Not sure how the microtubule polarity can be determined with tubulin labeling only.

6. Intriguingly, overexpression of Staufen caused other phenotypes besides the *osk* localization, such as failure of *bcd* accumulation in early oocytes (Figure S7F') and oocyte nucleus localization defects (Figure S7H-H'). Thus, it behaves more like a general dynein inhibitor, as both processes are known to be dynein-dependent. This is consistent with their in vitro assay with the mammalian BicD2N (Figures 2H and 2I). Thus, these GFP-Staufen overexpression phenotypes should be included in the discussion (Line 503-517).

Reviewer #2 (Comments to the Authors (Required)):

The authors have thoroughly addressed my previously stated concerns in this revised version and I can now recommend publication.

Reviewer #3 (Comments to the Authors (Required)):

The authors have been responsive to all comments with new analysis, changes to figures or acceptable rebuttal that additional experiments are beyond the scope of this study.

Response to Reviewer 1:

The authors added a significant amount of data and new analysis to support their conclusion. I do still have some remaining concerns, mostly regarding their presentation that are easy to fix.

We thank the reviewer for their thorough proofreading of our manuscript and for their suggestions about data presentation. We have highlighted the changes we have made in response to these suggestions and those requested by the Editor in blue font in the revised manuscript file.

(1) Overall, figures and supplementary figures should be referred to within the main text in numerical order. The authors jumped between the supplementary figures within the text, which is not reader-friendly. For example, at the beginning of the results: Line 99-101 "However, loss of Staufen does not seem to affect the overall RNA content of oskar RNPs in the oocyte as revealed by single-molecule fluorescent in situ hybridization (smFISH) (Fig S5B)". That is the first time they refer to any supplementary figure in the results, and it is Figure S5. There are numerous similar examples in the text.

We have now reorganized several figures so that the supplementary figures are introduced in numerical order in the text.

(2) In several cases, I was not able to locate panels that were mentioned in the main text. For example, in lines 210-212, the authors said "Further supporting this notion, removal of a functional copy of *egl* partially suppressed oskar mislocalization at the anterior of egg-chambers with reduced Staufen levels (*stau RNAi, egl1/+*; Fig S3C-E)". I was not able to locate the Figure S3C-E. Line 364-366: the authors write "However when we analyzed GFP-Dhc association with oskar RNPs, we found no major difference between *stau RNAi*, *Egl*-overexpressing, and control oocytes (Fig 6F and F')". Again, there is no Figure 6F'. Line 367-368: "Intriguingly, *stau RNAi* and *Egl* overexpressing oocytes also did not display an increase in the association of BicD with oskar RNPs (Fig 6G and G')". I cannot locate Figure 6G'. And Figure 6G is not about BicD association either. So where are the data about BicD association with osk RNPs?

We apologize for these errors:

- Fig. S3, C-E was mislabeled and should have been Fig. S4, C-E. In the revised manuscript, it is again Fig. S3, C-E, as the original Fig. S3 has been merged with Fig. S2 to conform to the request to reduce the number of supplementary figures.
- Fig. 6: In the previous submission we accidentally left out the data on BicD-GFP colocalization with *oskar* mRNPs. This has now been corrected, as has the issue around Fig. 6, F and F'.

(3) Figure legends need more information. I don't understand Figure S1. What are the blue, red, and green arrowheads? Is the GFP-tagged Staufen larger than the endogenous Staufen protein? What are the estimated kDa for Staufen and GFP-Staufen? Does the lane 3 Staufen-GFP refer to the one under the control of its endogenous *stau* promoter? Is the lane 4 GFP-Staufen overexpression referring to UASp-GFP-*stau* under *mat atub-Gal4* driver? All the information should be included in the figure legends. How the fraction in Figure 4F can be negative. Is it normalized to GFP-only control? Please include the related information in the legends. Another example is Figure 6A-C: how are BicD and p50 labeled? By UASp-BicD.eGFP and UAS-DCTN2-p50::GFP, respectively? If so, how are these transgenes driven? Why does *osk* mRNA have some mislocalization phenotype, especially in Figures 6A and 6B? All the information should be included in the Figure legends.

We have added the missing information to the figure legends as requested.

(4) Some figures have the error bars overlapping, making it really hard to distinguish, such as Figure 4E, Figure S5F, and Figure 6D-F. For Figure 4E, is there a statistically significant difference between control (blue) and *Bic1/+* (purple)?

We agree with the reviewer that in some instances it was difficult to interpret the graphs due to overlapping error bars. We have now introduced a slight offset in the x-axis between overlapping items in Figures 4-6 and S4 to address this issue. In general, we only tested whether the mean of the distributions have a true difference to zero ($\alpha = 0.01$, one sample t-test) but not the difference between conditions. In response to the reviewer's specific question, *Egl* association with *oskar* RNPs with ~4 copies of mRNA in control oocytes is not different from zero (indicated by triangles), whereas the same metric is significantly higher than zero in the *BicD[1]*+ oocytes.

Some minor scientific questions:

1. Line 391-392, the authors claimed that "both Staufen and Staufen-GFP signal accumulated in the oocyte even in the absence of oskar mRNA (Fig S7A-C)". However, the endogenous Staufen level is much lower in the oskar mutant background (Figure S7A) compared to the control (Figure S7A). It is not consistent with the text. And I cannot find any quantification of this set of data.

We did not quantify endogenous Staufen in these experiments, although this protein reproducibly accumulated in the oocyte in the absence of *oskar* mRNA, as we stated in the Results. We have performed quantitative analysis from multiple egg chambers of Staufen-GFP localization in control and *osk*[A87]- (RNA null) oocytes (Fig. S6A in the revised manuscript). This shows that there is only a modest reduction in accumulation of Staufen-GFP in *osk*[A87]- oocytes, and therefore this protein is still strongly enriched in the ooplasm. In contrast, there is very little enrichment of the protein in the ooplasm of *egl* RNAi egg chambers, which supports our model of an RNA-based delivery process.

2. What are the egg chamber stages quantified in Figure 7D and Figure S7D? Does the *stau* mRNA oocyte accumulation become less during mid-oogenesis (compared Figure 7E with Figure 7B)? If so, any quantification and explanation?

We quantified early (stage 4-6) egg-chambers for the analysis of *stau* mRNA accumulation; this is now indicated in the legends of Fig. 7 and Fig. S6. Because expansive growth of the oocyte during development dilutes the RNA in the ooplasm, we are not able to draw meaningful conclusions about relative enrichment of *stau* mRNA in the oocyte in early and mid-oogenesis stages.

3. What MCP-GFP line was used for the live imaging? I cannot find any information about the line: which promoter and which chromosome insertion? No reference either.

This information is now provided in the Methods section

4. Line 497-498: "dynein's association with oskar RNPs in the oocyte does not scale with RNA content". This is slightly misleading, as the authors showed that the Dhc amount does increase with the numbers of RNA per RNP (Figure 6E).

We have modified the text to clarify this issue (lines 358-363 of the Word document, 395-399 of the converted pdf):

"The amount of BicD associated with the RNPs in the oocyte was independent of *oskar* copy number. In contrast, p50 and Dhc association with oskar RNPs increased as a function of mRNA content, although not proportionally (~ two- to five-fold increase in signal from the respective proteins versus an ~100-fold increase in *oskar* mRNA content; Fig. 6 E). This result indicates that BicD, p50 and Dhc are not recruited to *oskar* RNPs by identical mechanisms."

5. In Figure 1, the *osk*-MS2 motility was analyzed in ooplasmic extracts. How was the movement direction determined? According to the Methods, these samples were labeled with mCherry-alpha-tubulin. Not sure how the microtubule polarity can be determined with tubulin labeling only.

We thank the reviewer for spotting this omission, which is now corrected in the Results (line 89 of the Word document, 120 of the converted pdf) and Methods (line 576 of the Word document, 618 of the converted pdf). Plus ends of dynamic microtubules in our ex vivo assays were marked with transgenically supplied EB1-mCherry (as reported previously; Gaspar et al., EMBO J 2017). This method highlights the last 3-5 microns of the growing microtubule plus tip, which after temporal averaging allows visualization of the last 10-15 microns of the microtubule.

6. Intriguingly, overexpression of Staufen caused other phenotypes besides the *osk* localization, such as failure of *bcd* accumulation in early oocytes (Figure S7F') and oocyte nucleus localization defects (Figure S7H-H'). Thus, it behaves more like a general dynein inhibitor, as both processes are known to be dynein-dependent. This is consistent with their in vitro assay with the mammalian BicD2N (Figures 2H and 2I). Thus, these GFP-Staufen overexpression phenotypes should be included in the discussion (Line 503-517).

We have incorporated this interesting point in the Results (lines 421-423 of the Word document, 459-461 of the converted pdf) and Discussion (lines 507-509 of the Word document, 548-550 of the converted pdf).